# INQUIRE: A Natural World Text-to-Image Retrieval Benchmark

**Edward Vendrow**[*1], **Omiros Pantazis**[*2], **Alexander Shepard**[3], **Gabriel Brostow**[2],
**Kate E. Jones**[2], **Oisin Mac Aodha**[†4], **Sara Beery**[†1], **Grant Van Horn**[†5]

[1]Massachusetts Institute of Technology    [2]University College London    [3]iNaturalist
[4]University of Edinburgh    [5]University of Massachusetts Amherst

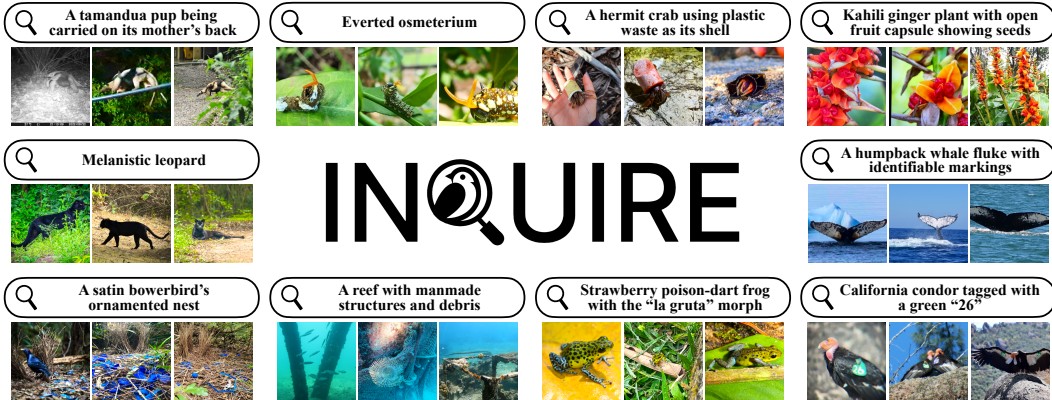

Figure 1: INQUIRE is a text-to-image retrieval benchmark of 250 expert-level queries comprehensively labeled over a new five million image dataset. The queries span a range of ecological and biodiversity concepts, requiring reasoning, image understanding, and domain expertise.

## Abstract

We introduce INQUIRE, a text-to-image retrieval benchmark designed to challenge multimodal vision-language models on expert-level queries. INQUIRE includes iNaturalist 2024 (iNat24), a new dataset of five million natural world images, along with 250 expert-level retrieval queries. These queries are paired with all relevant images comprehensively labeled within iNat24, comprising 33,000 total matches. Queries span categories such as species identification, context, behavior, and appearance, emphasizing tasks that require nuanced image understanding and domain expertise. Our benchmark evaluates two core retrieval tasks: (1) INQUIRE-FULLRANK, a full dataset ranking task, and (2) INQUIRE-RERANK, a reranking task for refining top-100 retrievals. Detailed evaluation of a range of recent multimodal models demonstrates that INQUIRE poses a significant challenge, with the best models failing to achieve an mAP@50 above 50%. In addition, we show that reranking with more powerful multimodal models can enhance retrieval performance, yet there remains a significant margin for improvement. By focusing on scientifically-motivated ecological challenges, INQUIRE aims to bridge the gap between AI capabilities and the needs of real-world scientific inquiry, encouraging the development of retrieval systems that can assist with accelerating ecological and biodiversity research.

---

*Equal contribution. †Equal supervision, order randomized.

38th Conference on Neural Information Processing Systems (NeurIPS 2024) Track on Datasets and Benchmarks.

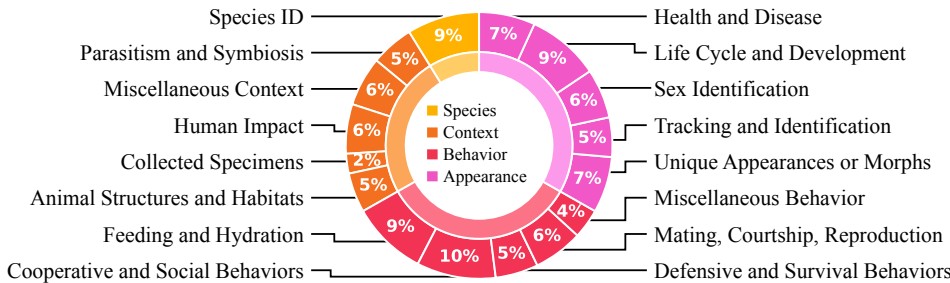

Figure 2: Category breakdown for the fine-grained queries that make up INQUIRE. Each query category falls under one of the following supercategories: Species, Context, Behavior, or Appearance.

# 1 Introduction

Recent advances in multimodal learning have resulted in advanced models [60; 43; 3] that demonstrate remarkable generalization capabilities in zero-shot classification [60; 83], visual question-answering (VQA) [39; 80; 4; 40], and image retrieval [80; 40]. These models offer the potential to assist in the exploration, organization, and extraction of knowledge from large image collections. However, despite this success, there remains a significant gap in the evaluation of these models on domain-specific, expert-level queries, where nuanced understanding and precise retrieval are critical. Addressing this gap is essential for future deployment in specialized fields such as biodiversity monitoring and biomedical imaging, among other scientific disciplines.

Previous studies of the multimodal capabilities of this new generation of models have primarily focused on the task of VQA. In VQA, it has been demonstrated that there remains a large performance gap between state-of-the-art models and human experts in the context of challenging perception and reasoning queries such as those found on college-level exams [81; 84]. However, no such expert-level benchmark exists for *image retrieval*. The most commonly used text-to-image retrieval benchmarks are derived from image captioning datasets, and contain simple queries related to common everyday categories [79; 42]. Current multimodal models achieve near perfect performance on some of these benchmarks, indicating that they no longer pose a challenge (e.g., BLIP-2 [40] scores 98.9 on Flickr30K [79] top-10). Existing retrieval datasets are generally small [58; 59; 79; 42], limited to a single visual reasoning task (e.g., landmark-location matching [58; 59; 74]), and lack concepts that would require expert knowledge [58; 59; 74; 79; 42]. These limitations impede our ability to track and improve image retrieval capabilities.

A domain that is well-suited for studying this problem is the natural world, where images collected by enthusiast volunteers provide vast and largely uncurated sources of publicly available scientific data. In particular, the iNaturalist [2] platform contains over 180 million species images and contributes immensely to research in biodiversity monitoring [16; 48]. These images also contain a wealth of "secondary data" not reflected in their species labels [57], including crucial insights into interactions, behavior, morphology, and habitat that could be uncovered through searches. However, the time-consuming and expert-dependent analysis needed to extract such information prevents scientists from taking advantage of this valuable data at scale. This cost is amplified as scientists typically want to retrieve *multiple* relevant images for each text query, so that they can track changes of a property over space and time [78]. This domain serves as an ideal testbed for expert image retrieval, as these images contain expert-level diverse and composite visual reasoning problems, and progress in this field will enhance impactful scientific discovery.

In this work, we introduce INQUIRE, a new dataset and benchmark for expert-level text-to-image retrieval and reranking on natural world images. INQUIRE includes the iNat24 dataset and 250 ecologically motivated retrieval queries. The queries span 33,000 true-positive matches, pairing each text query with all relevant images that we comprehensively labeled among iNat24's five million natural world images. iNat24 is sampled from iNaturalist [2], and contains images from 10,000 different species collected and annotated by citizen scientists, providing significantly more data for researchers interested in fine-grained species classification. The queries contained within INQUIRE come from discussions and interviews with a range of experts including ecologists, biologists, ornithologists, entomologists, oceanographers, and forestry experts.

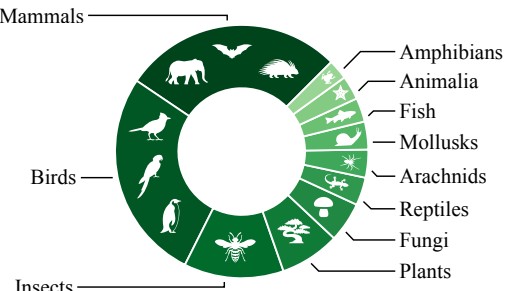

Figure 3: Proportion of queries in INQUIRE associated with each iconic group of species.

Table 1: Comparison to common datasets used to evaluate text-to-image retrieval [26]. Unlike other datasets, INQUIRE has significantly more images and many matches per query rather than exactly one. *MpQ: Matches per query*

| Dataset | Images | Queries | MpQ | Expert |
|---|---|---|---|---|
| Flickr30k [79] | 1,000 | 5k | 1 | ✗ |
| COCO [42] | 5,000 | 25k | 1 | ✗ |
| **INQUIRE** | 5,000,000 | 250 | 1–1.5k | ✓ |

Our evaluation of multimodal retrieval methods demonstrates that INQUIRE poses a significant challenge, necessitating the development of models able to perform expert-level retrieval within large image collections. A key finding from our experiments is that reranking, a technique typically used in text retrieval [54; 35; 34], offers a promising avenue for improvement in image retrieval. We hope that INQUIRE will inspire the community to build next-generation image retrieval methods towards the ultimate goal of accelerating scientific discovery. We make INQUIRE, the iNat24 dataset, pre-computed outputs from state-of-the-art models, and code for evaluation available at https://inquire-benchmark.github.io/.

## 2 Related Work

**Vision-Language Models (VLMs).** Large web-sourced datasets containing paired text and images have enabled recent advances in powerful VLMs [15; 85]. Contrastive methods such as CLIP [60] and ALIGN [32], among others, learn an embedding space where the data from the two modalities can be encoded jointly. The ability to reason using natural language and images together has yielded impressive results in a variety of text-based visual tasks such as zero-shot classification [60; 83], image captioning [39; 80; 4; 30; 40], and text-to-image generation [53; 7; 61; 64; 8]. However, the effectiveness of these contrastive VLMs for more complex compositional reasoning is bottlenecked by the information loss induced by their text encoders [33].

There also exists a family of more computationally expensive VLMs that connect the outputs of visual encoders directly into language models. Models like LLaVA [43; 44], BLIP [39; 40; 21], and GPT-4o [3; 55] have demonstrated impressive vision-language understanding. However, despite their potential for answering complex vision-language queries, these models are not suitable for processing large sets of images at interactive rates, which is essential for retrieval, due to their large computational requirements during inference. In this paper, we do not introduce new VLMs, but aim to better understand the capabilities and shortfalls of existing methods for text-to-image retrieval.

**Image Retrieval.** Effective feature representations are essential for achieving strong image retrieval performance. Earlier approaches from image-to-image used hand-crafted features [49; 12] but these have largely been replaced with deep learning-based alternatives [36; 9; 6; 11]. More recently, in the context of text-to-image retrieval, we have seen the adoption of contrastive VLMs [60; 32] trained on web-sourced paired text and image datasets. These models enable zero-shot text-based retrieval and have been demonstrated to exhibit desirable scaling properties as training sets become larger [26; 24]. However, despite the potential of VLMs for image retrieval, their evaluation has been mostly limited to small datasets adapted from existing image captioning benchmarks, such as Flickr30k [79] and COCO [42], which contain just 1,000 and 5,000 images, respectively. Furthermore, recent models are saturating performance on these less challenging datasets, e.g., BLIP-2 [40] scores 98.9 on Flickr30K and 92.6 on COCO top-10 text-to-image retrieval. As most text-to-image benchmarks have been derived from image captioning datasets, each query is a descriptive caption that matches exactly one image. In contrast, real-world retrievals often involve multiple images relevant to a single query, and the query itself typically does not describe every aspect of the images as thoroughly as a caption does. We compare INQUIRE to common text-to-image retrieval datasets in Table 1.

More recent datasets have been purpose-built to probe specific weaknesses of retrieval systems, such as compositionality [50; 29; 62], object relationships [82], negation [72; 66], and semantic

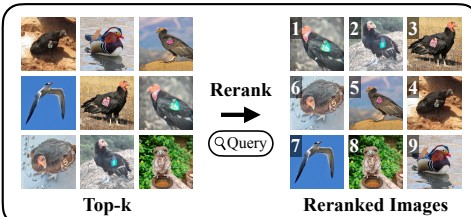

Figure 4: The INQUIRE benchmark consists of a full-dataset ranking task and a reranking task targeting different aspects of the image retrieval problem.

granularity [76]. [25] created a retrieval dataset for camera trap images, but use image captions that were automatically generated from a small set of discrete image attributes, limiting their utility beyond this set. The problem of fine-grained retrieval, where there may only be subtle visual differences between concepts of interest, has also been explored extensively [73]. However, typically these datasets convert existing classification datasets to the retrieval setting, resulting in small image pools and limited query diversity. INQUIRE addresses these shortcomings with a considerably larger image set and fine-grained queries that require advanced image understanding and domain expertise.

**Reranking.** In text retrieval, a common workflow is to first efficiently obtain an initial ranking of documents using pre-computed text embeddings and then *rerank* the top retrievals with a more costly but sophisticated model [54; 35; 34]. While VLMs like CLIP [60] enable efficient image retrieval and more expensive models such as GPT-4o [55] could perform more complex ranking, this workflow has not been extensively explored in text-to-image applications primarily due to a lack of evaluation datasets. To this end, INQUIRE introduces a reranking challenge to drive further progress on this task.

**Expert-level Benchmarks.** Visual classification benchmarks have evolved from simply containing common everyday categories [63; 42] to having more "expert-level" concepts [68; 69]. Challenging datasets like the iNaturalist benchmarks [69; 70], contain large class imbalances and fine-grained concepts that require expert-level knowledge to identify. The NeWT benchmark from [70] is similar in spirit to INQUIRE in that it proposes a collection of natural world questions. However, NeWT is a set of binary classification challenges, and while there is a variety of tasks, the majority of them are standard species classification. Further, NeWT uses a small (200–400) fixed set of positive and negative labeled images for each task, so it is not suitable for evaluating retrieval.

In general, evaluation benchmarks have struggled to keep pace with the growing capabilities of recent large models which perform very well on them [5]. For language, specific datasets have been developed to challenge common sense reasoning abilities [52; 14]. Multimodal datasets have also been proposed to assess vision-language capabilities [75; 38; 46; 77]. Nevertheless, these benchmarks test general skills in tasks that are not particularly challenging for humans and thus, are not testing a models' abilities in scenarios where expert-level knowledge is required.

To address the need for more difficult benchmarks, recent expert-level benchmarks have been devised for LLMs [28; 84] and multimodal models [81; 51]. For instance, MMMU [81] features questions that cover a range of college-level disciplines while Encyclopedic-VQA [51] comprises visual questions related to fine-grained entities which demand encyclopedic knowledge. The relatively low performance on these benchmarks, compared to human performance, highlights current limitations in multimodal models. However, there is no equivalent expert-level dataset for fine-grained text-to-image retrieval. INQUIRE fills this gap by providing a set of challenging and visually fine-grained retrieval questions focused on real-world tasks in retrieval from natural world image collections.

## 3 The INQUIRE Benchmark

Here we describe INQUIRE, our novel benchmark for assessing expert-level image retrieval for fine-grained queries on natural world image collections. INQUIRE consists of a collection of 250 queries, where each query is represented as a brief text description of the concept of interest (e.g., *"Alligator lizards mating"* [56]), and contains its relevant image matches comprehensively labeled over a dataset of five million natural world images. These queries represent real scientific use cases collected to cover diverse, expert sources including discussions with scientists across environmental and ecological

disciplines. Several examples of our queries are illustrated in Figure 1, with more in Appendix J. Our queries challenge retrieval methods to demonstrate fine-grained detail recognition, compositional reasoning, character recognition, scene understanding, or natural world domain knowledge. While queries can require expert-level knowledge, the information needed to solve them is publicly available online and thus feasible for large web-trained models to learn. In this section, we detail the data sources utilized for the construction of INQUIRE, describe the data collection process, and introduce two image retrieval tasks — INQUIRE-FULLRANK and INQUIRE-RERANK — that address different aspects of real-world text-to-image retrieval.

## 3.1 The iNaturalist 2024 Dataset

As part of the INQUIRE benchmark, we create a new image dataset, which we refer to as iNaturalist 2024 (iNat24). This dataset contains five million images spanning 10,000 species classes collected and annotated by community scientists from 2021–2024 on the iNaturalist platform [2]. iNat24 forms one of the largest publicly available natural world image repositories, with twice as many images as in iNat21 [70]. To ensure cross-compatibility for researchers interested in using both datasets, iNat24 and iNat21 have the same classes but do not contain the same images, freeing iNat21 to be used as a training set. The sampling and collection process of iNat24 is in Appendix H.

## 3.2 Query and Image Collection Process

**Query Collection.** To ensure that INQUIRE comprises text queries that are relevant to scientists, we conducted interviews with individuals across different ecological and environmental domains - including experts in ornithology, marine biology, entomology, and forestry. Further queries were sourced from reviews of academic literature in ecology [57]. Representative queries and statistics can be seen in Figures 1, 2, and 3. We retained only queries that (1) could be discerned from images alone, (2) were feasible to comprehensively label over the entire iNat24 dataset, and (3) were of interest to domain experts.

**Image Annotation.** All image annotations were performed by a small set of individuals whose interest and familiarity with wildlife image collections enabled them to provide accurate labels for challenging queries. Annotators were instructed to label all candidate images as either *relevant* (i.e., positive match) or *not relevant* (i.e., negative match) to a query, and to mark an image as not relevant if there was reasonable doubt. To allow for comprehensive labeling, where applicable, iNat24 species labels were used to narrow down the search to a sufficiently small size to label all relevant images for the query of interest. For queries in which species labels could not be used, labeling was performed over the top CLIP ViT-H-14 [24] retrievals alone. In this case, the resulting annotations were only kept if we were certain that this labeling captured the vast majority of positives, including labeling until at least 100 consecutive retrievals were not relevant (see Appendix H). Queries that were deemed too easy, not comprehensively labeled, or otherwise not possible to label were excluded from our benchmark. In total, this process resulted in 250 queries which involved labeling 194,334 images, of which 32,696 were relevant to their query. Further details are in Appendix H.

**Query Categories.** Each query belongs to one of four supercategories (appearance, behavior, context, or species), and further into one of sixteen fine-grained categories (e.g., Animal Structures and Habitats). Figure 2 shows the distribution of query categories, and Figure 3 shows the distribution of iconic groups of the species represented by each query (e.g., Mammals, Birds). We also note queries that use scientific terminology, words typically used only within scientific contexts (e.g., "A godwit performing distal rhynchokinesis").

**Data Split.** We divide all queries into 50 validation and 200 test queries using a random split, stratified by category.

## 3.3 Retrieval Tasks

We introduce two tasks to address different aspects of the text-to-image retrieval problem. Real-world retrieval implementations often consist of two stages: an initial top-k retrieval with a more computationally efficient method (e.g., CLIP zero-shot using pre-computed image embeddings), followed by a reranking of the top-k retrievals with a more expensive model. To enable researchers to explore both stages, while ensuring that those with more limited computational resources can participate, we follow previous large-scale reranking challenges like TREC [19; 20] by offering both a full dataset retrieval task and a reranking task (see Figure 4).

Table 2: INQUIRE-FULLRANK retrieval performance for selected CLIP-style models. Larger models, trained on higher quality datasets, tend to achieve better performance.

| Training dataset | Method | Params (M) | mAP@50 | nDCG@50 | MRR |
|---|---|---|---|---|---|
| WildCLIP [25] | CLIP ViT-B-16 | 150 | 7.4 | 16.1 | 0.33 |
| BioCLIP [67] | CLIP ViT-B-16 | 150 | 5.0 | 8.6 | 0.17 |
| OpenAI [60] | CLIP RN50 | 102 | 6.8 | 15.1 | 0.29 |
| | CLIP RN50x16 | 291 | 13.6 | 25.5 | 0.48 |
| | CLIP ViT-B-32 | 151 | 7.5 | 16.8 | 0.30 |
| | CLIP ViT-B-16 | 150 | 10.4 | 20.9 | 0.40 |
| | CLIP ViT-L-14 | 428 | 14.4 | 27.1 | 0.46 |
| DFN [24] | CLIP ViT-B-16 | 150 | 15.1 | 28.1 | 0.48 |
| | CLIP ViT-L-14 | 428 | 23.1 | 37.3 | 0.54 |
| | CLIP ViT-H-14@378 | 987 | 33.3 | 48.8 | **0.69** |
| WebLI [83] | SigLIP ViT-L-16@384 | 652 | 31.1 | 46.6 | 0.68 |
| | SigLIP SO400m-14@384 | 878 | **34.2** | **49.1** | **0.69** |

**INQUIRE-FULLRANK.** The goal of this task is end-to-end retrieval, starting from the entire five million image iNat24 dataset. Progress on the full retrieval task can be made with better and more efficient ways to organize, process, filter, and search large image datasets. Although performance will increase with improvements to either of the two stages in a typical retrieval pipeline, we hope this task also encourages the development of retrieval systems beyond the two-stage approach.

**INQUIRE-RERANK.** This task evaluates reranking performance from a fixed initial ranking of 100 images. We believe that significant progress in retrieval will come from developing better reranking methods that re-order an initial retrieved subset. Thus, fixing the starting images for each query provides a consistent evaluation of reranking methods. This task also lowers the barrier to entry by giving researchers a considerably smaller set of top retrievals to work with, rather than requiring them to implement an end-to-end retrieval system. The top 100 ranked images for each query are retrieved using CLIP ViT-H-14 zero-shot retrieval on the entire iNat24 dataset. Consistent with previous large-scale reranking challenges [19; 20; 37], we retain only queries for which at least one positive image is among the top 100 retrieved images and no more than 50% of these top images are relevant. This ensures that the reranking evaluation remains meaningful and discriminative. This filtering process yields a task subset of 200 queries (reduced from our original 250 queries), split into 40 validation and 160 test queries according to the original validation/test split, with and 4,000 and 16,000 corresponding images, respectively.

## 4 Retrieval Methods

The goal of text-to-image retrieval is to rank images from a potentially large image collection according to their relevance to an input text query. Here, we describe the retrieval and reranking methods that we evaluate, covering current state-of-the-art approaches.

**Embedding Similarity.** Models such as CLIP [60] are well suited for the text-to-image retrieval setting as they operate on a joint vision and language embedding space. In this setting, similarity between an image and text query is simply determined by their cosine similarity. The key advantage of embedding models is that the embedding for each image can be pre-computed once offline as they do not change over time. At inference time, only the embedding of the text query needs to be computed and then compared to the cached image embeddings for retrieval. This is helpful as the number of images we wish to search over can be on the order of millions, or even billions [65]. Thus to speed up retrieval, the image embeddings can be pre-computed and indexed using approximate nearest neighbor methods [23], allowing for near-instantaneous retrievals on large collections. This is beneficial both for end-to-end retrieval and as the first step for a multi-stage retrieval approach. We also benchmark recent models such as WildCLIP [25] and BioCLIP [67] which are adapted versions of CLIP that explicitly target natural world use cases.

**Reranking with Multimodal Models.** Reranking is a common paradigm in text retrieval, where a rapid search through pre-computed document indexes for potential matches is followed by a more expensive reranking of the top retrievals [54; 35; 34]. In the image domain, reranking has been

Table 3: Results for the INQUIRE-FULLRANK task using two-stage retrieval. The top-k images are retrieved with CLIP ViT-H/14 and then reranked with the selected large multimodal models. Reranking offers a significant avenue of improvement.

| Method | Rerank Top 50 | | | Rerank Top 100 | | |
|---|---|---|---|---|---|---|
| | mAP@50 | nDCG@50 | MRR | mAP@50 | nDCG@50 | MRR |
| *Initial ranking (ViT-H/14)* | 33.3 | 48.8 | 0.69 | 33.3 | 48.8 | 0.69 |
| *Best possible rerank* | 50.3 | 60.2 | 0.94 | 65.6 | 72.7 | 0.96 |
| *Open-source multimodal models* | | | | | | |
| BLIP-2 FLAN-T5-XXL [40] | 32.7 | 47.7 | 0.62 | 31.2 | 46.5 | 0.58 |
| InstructBLIP-T5-XXL [21] | 34.1 | 49.0 | 0.67 | 33.0 | 48.3 | 0.64 |
| PaliGemma-3B-mix-448 [13] | 35.0 | 49.7 | 0.70 | 35.6 | 50.6 | 0.68 |
| LLaVA-1.5-13B [44] | 33.1 | 48.4 | 0.66 | 32.2 | 47.9 | 0.64 |
| LLaVA-v1.6-7B [45] | 33.3 | 48.4 | 0.66 | 32.3 | 47.9 | 0.62 |
| LLaVA-v1.6-34B [45] | 34.8 | 49.7 | 0.69 | 35.7 | 51.2 | 0.69 |
| VILA-13B [41] | 35.0 | 49.6 | 0.67 | 35.7 | 50.8 | 0.65 |
| VILA-40B [41] | **37.4** | **51.4** | **0.73** | **40.2** | **54.6** | **0.72** |
| *Proprietary multimodal models* | | | | | | |
| GPT-4V [3] | 35.8 | 50.7 | 0.73 | 36.5 | 51.9 | 0.72 |
| GPT-4o [55] | **39.6** | **53.4** | **0.79** | **43.7** | **57.9** | **0.78** |

comparatively rare as the types of datasets for which it can be used are limited. In our experiments, we show that multimodal language models such as LLaVA [45], VILA [41], and GPT-4 [3; 55] are effective rerankers out-of-the-box. To adapt these multimodal models for ranking, which requires a continuous score for a given text query and image pair, we prompt: *Does this image show {some query}? Answer with "Yes" or "No" and nothing else.* (precise prompting details used for each model can be found in Appendix I). The logits of the "Yes" and "No" tokens are then used to compute the score: $s = s_y/(s_y + s_n)$, where $s_y = \exp(logit_{Yes})$ and $s_n = \exp(logit_{No})$.

# 5 Results

Here we present a comprehensive evaluation of retrieval methods on INQUIRE. All results are reported on the test set. Additional results, including on the validation set, are in Appendix E.

## 5.1 Metrics

We evaluate using Average Precision at k (AP@k), Normalized Discounted Cumulative Gain (nDCG), and Mean Reciprocal Rank (MRR). We primarily discuss AP as we find that this metric is the most discriminative of model performance. While these metrics have been commonly used to evaluate text retrieval, especially in the context of large-scale document retrieval [71; 19], they have not found use in image retrieval due to the nonexistence of benchmarks like INQUIRE containing many relevant images for retrieval, rather than just one. Thus, we include them in our analysis to encourage their use in future image retrieval research. We note that the utilized AP@k metric uses a modified normalization factor suited to the retrieval setting.

Existing image retrieval benchmarks typically evaluate using the recall@k metric (e.g., [40]), measuring if any of the top k images are relevant. While this makes sense in the setting where just one image is relevant, INQUIRE has potentially many relevant images and thus, we employ metrics that measure both relevance and ranking of retrievals. Detailed discussion our metrics is provided in Appendix G.

## 5.2 Fullrank Retrieval Task Results

We report full retrieval evaluation on INQUIRE in Tables 2 and 3. The per-category performance of selected CLIP models is reported in Figures 5 and 6. Further detailed results are in Appendix E.

**The best CLIP models leave significant room for improvement.** Table 2 shows that the top performing CLIP model achieves a moderate mAP@50 of 35.6. Although scaling models increases

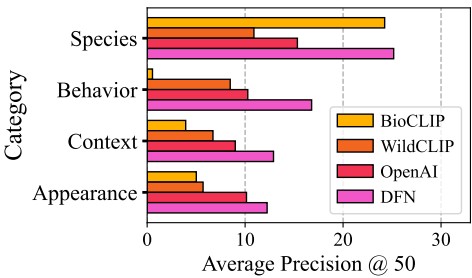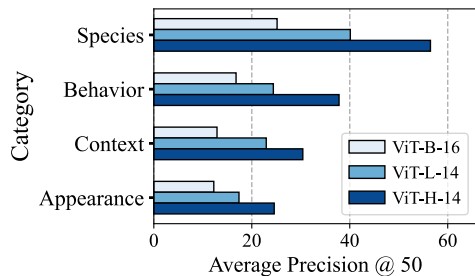

Figure 5: **Left:** CLIP zero-shot retrieval performance across supercategories using an identical backbone (ViT-B/16) trained or fine-tuned on different datasets. We see how training datasets have a significant effect on final performance, e.g., BioCLIP is tuned on natural world data at the expense of forgetting other categories. **Right:** CLIP retrieval performance of models trained on DFN [24].

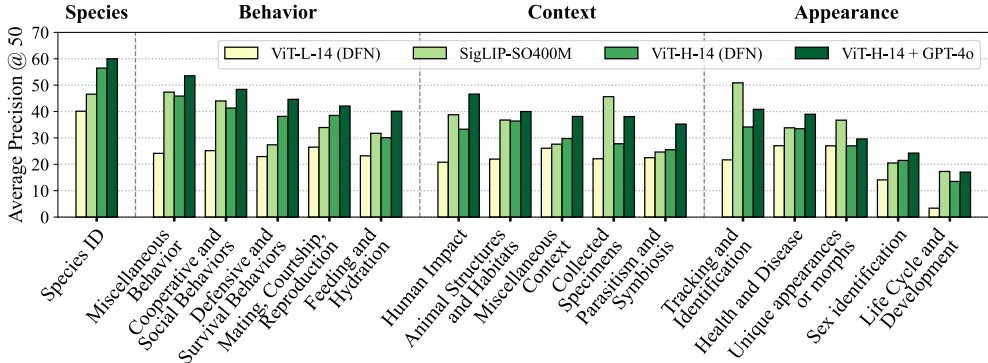

Figure 6: Retrieval performance for selected methods on INQUIRE query categories, ordered by difficulty. Some categories like LIFE CYCLE AND DEVELOPMENT are exceptionally hard for current models. GPT-4o reranking improves performance in every category over its initial ViT-H-14 ranking.

performance (see Figure 5), these results suggest that just scaling might not be enough, so future research should seek methods to better incorporate domain knowledge.

**Small models struggle to answer many queries.** In Table 2 we can see that CLIP RN50 and CLIP ViT-B-32 score an mAP@50 of just 7.6 and 8.2 respectively, demonstrating that these smaller models are unable to provide accurate retrievals for nearly all queries. Since the largest models get comparatively much higher scores, the queries are not impossible but rather difficult for smaller models. DFN ViT-B-16, trained with curated data, outperforms the larger OpenAI ViT-L-14, emphasizing the opportunity to improve the performance of efficient models via better data or training methods.

**High-quality training data is crucial for expert-level queries**. In Figure 5-left we show the retrieval performance on different supercategories for CLIP ViT-B/16 models that are trained on different datasets: BioCLIP [67], WildCLIP [25], OpenAI [60], and DFN [24]. The DFN model, trained on two billion filtered image-text pairs, is the best generalist model on OpenCLIP's benchmarks [31] and also outperforms all the others here, demonstrating the effectiveness of high quality pretraining. Conversely, models specifically trained on natural world data demonstrate degraded performance: BioCLIP was trained primarily on taxonomic captions and images, including iNat21, yet fails significantly on non-species queries, while WildCLIP has degraded performance in all supercategories. This performance emphasizes the need for better natural world models and fine-tuning strategies that can gain domain-specific expertise while preserving generalist capabilities.

**Reranking offers a valuable opportunity for improving retrieval.** Table 3 shows that reranking with larger models like VILA-40B and GPT-4o gives a significant performance boost in mAP@50 of 7 and 12 points, respectively. Still, even GPT-4o performs significantly worse than the best possible rerank of its initial CLIP ViT-H-14 ranking. Increasing the size of the initial retrieval set from 50 to 100 can further improve performance by surfacing more relevant images, but only higher-performing models benefit: The mAP@50 for GPT-4o increases by 5 points, while lower-performing models

Table 4: Results for the INQUIRE-RERANK task on various embedding and multimodal models. For each task, a fixed set of the top-100 images is provided, which we then rerank using different methods. Evaluation metrics are calculated based solely on this fixed set, disregarding any potential positives outside of the top-100 images. Therefore, a perfect score is achievable within this context.

| Method | AP | nDCG | MRR | Method | AP | nDCG | MRR |
|---|---|---|---|---|---|---|---|
| *Random* | 22.1 | 52.6 | 0.35 | | | | |
| | | | | *Open-source multimodal models* | | | |
| *Embedding models* | | | | BLIP-2 T5-XXL [40] | 40.0 | 65.4 | 0.55 |
| CLIP ViT-B-32 [60] | 30.2 | 59.1 | 0.47 | InstructBLIP-T5-XXL [21] | 41.5 | 66.9 | 0.59 |
| CLIP ViT-L-14 [60] | 36.8 | 64.2 | 0.57 | PaliGemma-3B-mix-448 [13] | 42.9 | 67.9 | 0.60 |
| CLIP ViT-H-14 [24] | 42.6 | 68.7 | 0.66 | LLaVA-v1.5-13B [44] | 43.7 | 68.4 | 0.61 |
| SigLIP SO400m-14 [83] | **50.1** | **73.5** | **0.72** | LLaVA-v1.6-7B [45] | 46.9 | 70.4 | 0.66 |
| | | | | LLaVA-v1.6-34B [45] | 47.0 | 70.4 | 0.62 |
| *Proprietary multimodal models* | | | | VILA-13B [41] | 47.1 | 71.1 | 0.67 |
| GPT-4V [3] | 47.8 | 71.9 | 0.70 | VILA-40B [41] | **52.8** | **74.4** | **0.71** |
| GPT-4o [55] | **59.6** | **78.9** | **0.78** | | | | |

like LLaVA-v1.6-7B see decreased performance. Further results for varying the initial ranking set size are in Appendix E. Figure 6 visualizes how GPT-4o reranking improves performance on every category compared to its initial ViT-H-14 ranking.

**Different query types present challenges of varying difficulties to existing models.** Figure 6 illustrates the difference in performance across query categories. We see that APPEARANCE queries, which often require both domain knowledge of an organism's appearance and the fine-grained visual reasoning to recognize them, are the most difficult for existing models. Indeed, the LIFE CYCLE AND DEVELOPMENT set (e.g., *"Immature bald eagle"*, *"A cicada in the process of shedding its exoskeleton"*) are by far the most difficult. Conversely, CONTEXT queries such those in the HUMAN IMPACT set (e.g., *"leopard on a road"*, *"bird caught in a net"*), for which less expertise and comparatively coarser image understanding are needed, are easier for existing models.

### 5.3 Rerank Retrieval Task Results

The results for the INQUIRE-RERANK task are presented in Table 4, where we evaluate reranking performance of both CLIP-style models like ViT-B-32 and larger vision-language models such as GPT-4o. Since the total number of images for each query is small (i.e., 100), we also show the expected results of a random reranking for baseline comparison. In Table 5 we further break down INQUIRE-RERANK results by queries containing scientific terminology and by query supercategory.

**Current models struggle with expert-level text-to-image retrieval on INQUIRE.** In Table 4 we observe that the highest average precision score of 59.6, achieved by GPT-4o, is far below the perfect score of 100, showing substantial room for improvement. Smaller models like CLIP ViT-B-32 only slightly outperform random chance. Since the top retrieved are often visually or semantically similar, lower-performing models may be confused into promoting irrelevant images, leading to poorer ranking.

**Queries with scientific terminology are significantly more challenging, showing that models might not understand domains-specific language**. For example, the query *"Axanthism in a green frog"*—referring to a mutation limiting yellow pigment production, resulting in a blue appearance— uses specialized terminology that a model may not understand. As a result, a model may incorrectly rank typical green frogs higher than axanthic green frogs, leading to worse-than-random performance. We show the performance of reranking models on queries with scientific terminology in Table 5. Interestingly, GPT-4o appears to be closing this gap, with an average difference of 7 points between queries with and without scientific terminology (AP scores of 53 and 60, respectively), compared to a 16-point difference for the next best model, VILA-40B (AP of 39 and 55). Nevertheless, this gap remains. Future work should explore methods to improve models' comprehension of domain-specific language, which is critical for accurate retrieval in scientific contexts.

**Reranking effectiveness varies widely by the query type.** Table 5 shows that CONTEXT queries, often requiring general visual understanding, benefit substantially from reranking. Conversely, SPECIES queries, requiring fine-grained visual understanding, see minimal improvement, with the

Table 5: Evaluation of INQUIRE-RERANK with queries grouped into different query types. First, we group queries containing scientific lingo and no scientific lingo. Next, we group queries by their supercategory (Appearance, Behavior, Context, Species). Queries with lingo tend to be more difficult, especially for large models with good generalist understanding but lacking domain expertise. All results are reported in AP.

| | By Lingo | | By Supercategory | | | |
|---|---|---|---|---|---|---|
| Model | Lingo | No Lingo | Appearance | Behavior | Context | Species |
| WildCLIP | 20.8 | 32.0 | 31.8 | 29.8 | 35.7 | 36.0 |
| BioCLIP | 17.2 | 30.3 | 27.8 | 25.6 | 31.3 | **44.8** |
| CLIP ViT-B-32 | 22.8 | 31.6 | 29.5 | 31.0 | 36.0 | 35.3 |
| CLIP ViT-L-14 | 29.2 | 37.4 | 37.2 | 36.2 | 40.2 | 38.2 |
| CLIP ViT-H-14 | 32.1 | 44.0 | 38.1 | 50.5 | 45.5 | 31.2 |
| SigLIP SO400m-14 | **38.3** | **51.7** | **51.7** | **53.6** | **54.1** | 44.6 |
| LLaVA-v1.6-34B | 28.2 | 49.5 | 41.0 | 48.2 | 53.8 | **37.5** |
| VILA-13B | 37.2 | 48.2 | 39.3 | 47.1 | 58.6 | 34.8 |
| VILA-40B | **38.6** | **54.5** | **46.9** | **54.9** | **63.1** | 37.0 |
| GPT-4V | 35.9 | 49.3 | 40.3 | 50.2 | 54.2 | 39.0 |
| GPT-4o | **53.3** | **60.4** | **51.9** | **61.4** | **75.4** | **44.3** |

specialized BioCLIP beating out even GPT-4o. These trends suggest that while recent models have better generalized vision capabilities, they continue to struggle with fine-grained visual understanding.

## 6 Limitations and Societal Impact

While the species labels for each image in iNat24 are generated via consensus from multiple citizen scientists, there may still be errors in the labels which our evaluation will inherit. However, this error rate is estimated to be low [47]. INQUIRE contains natural world images, which while diverse, may hinder the relevance of some of our insights to other visual domains. In spite of this, we believe that due to the wide range of visual queries contained within, progress on INQUIRE will likely be indicative of multimodal model performance on other challenging domains.

There could be unintended negative consequences if conservation assessments were made based on the predictions from biased or inaccurate models evaluated in this paper. Where relevant, we have attempted to flag these performance deficiencies. While we have filtered out personally identifiable information from our images, the retrieval paradigm allows for free-form text search and thus care should be taken to ensure that appropriate text filters are in-place to prevent inaccurate or hurtful associations being made between user queries and images of wildlife.

## 7 Conclusion

We introduced INQUIRE, a challenging new text-to-image retrieval benchmark which consists of expert-level text queries that have been exhaustively annotated across a large pool of five million natural world images called iNat24. This benchmark aims to emulate real world image retrieval and analysis problems faced by scientists working with these types of large-scale image collections. Our hope is that progress on INQUIRE will drive advancements in the real scientific utility of AI systems. Our evaluation of existing methods reveals that INQUIRE poses a significant challenge even for the current largest state-of-the-art multimodal models, showing there is significant room for innovations to develop accurate retrieval systems for complex visual domains.

## Acknowledgments and Disclosure of Funding

We wish to thank the many iNaturalist participants for continuing to share their data and also the numerous individuals who provided suggestions for search queries. Special thanks to Kayleigh Neil, Beñat Yañez Iturbe-Ormaeche, Filip Dorm, and Patricia Mrazek for data annotation. Funding for annotation was provided by the Generative AI Laboratory (GAIL) at the University of Edinburgh. EV and SB were supported in part by the Global Center on AI and Biodiversity Change (NSF OISE-2330423 and NSERC 585136). OMA was in part supported by a Royal Society Research Grant. OP and KJ were supported by the Biome Health Project funded by WWF-UK.

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
