# Appendix

> Please refer to the links below for the dataset, repo, website, and live query demo:
>
> - Website: https://inquire-benchmark.github.io/
> - GitHub: https://github.com/inquire-benchmark/INQUIRE
> - Data: https://github.com/inquire-benchmark/INQUIRE/tree/main/data
> - Live Query Demo: http://ec2-3-147-61-23.us-east-2.compute.amazonaws.com/demo

# Contents

# A   INQUIRE Query Examples

Below we include several queries from INQUIRE with their broad justification, small number of examples of relevant and not relevant images, and a detailed explanation of each image's relevance.

🔍 California Condor tagged with green 26

California condors were extinct in the wild before their re-introduction in 1992. To track the movements of individuals, each condor has identifiable tags placed by biologists. Identifying individual condors based on community volunteer photos can provide evidence of their movement, behavior, and social interactions.

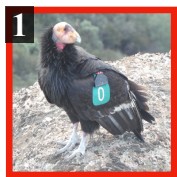 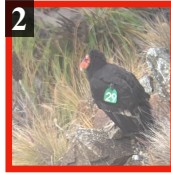 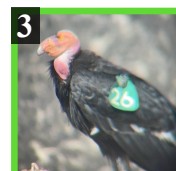 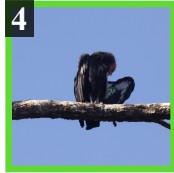 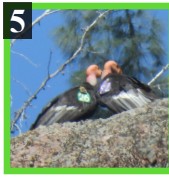

**Explanation**: (1) and (2) have green tags with numbers 0 and 29, respectively. (3), (4), and (5) all show a green tag 26, indicating that these are relevant.

🔍 Moorish Gecko with regenerated tail

This query is useful for studying tail autotomy and regeneration in geckos. Moorish geckos can regenerate their tail, but the regenerated tail will not be the same as the original: they do not grow tubercles so they appear smooth instead of ridged.

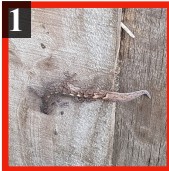 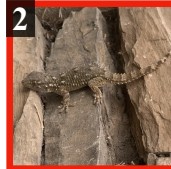 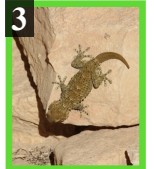 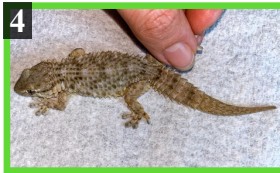 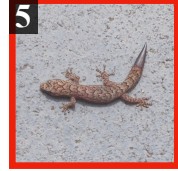

**Explanation**: (1) and (5) have regenerated tails, but are not Moorish geckos. (2) is a Moorish gecko, but does not have a regenerated tail as evidenced by ridges all the way down the length of the tail. Finally, (3) and (4) are both Moorish geckos and have sections of their tails that appear entirely smooth, indicating that they are regenerated

🔍 Everted osmeterium

Swallowtail butterfly (Papilionidae) larvae have osmeterium, a unique defensive organ that is everted in response to threats. This organ has a few defensive uses: it secretes an acidic mixture that can deter threats, mimics a forked tongue to perhaps appear like a snake, and is brightly colored as a possible aposematic warning.

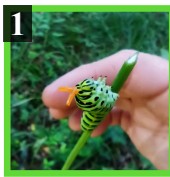 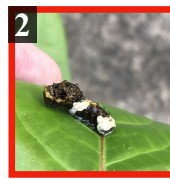 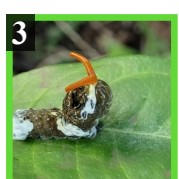 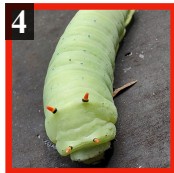 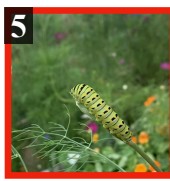

**Explanation**: (1) and (3) are correctly retrieved examples of swallowtail larvae at different life stages with everted osmeterium. (2) and (5) are also swallowtail larvae, but their osmeterium are not everted. (4) is a tulip-tree silk moth, which has four orange-red spurs on its head that are not osmeterium.

## 🔍 A godwit performing distal rhynchokinesis

Distal rhynchokinesis is an ability possessed by some long-billed shorebirds characterized by bending of the upper mandible. While it is hypothesized that this ability can help capture more feed when the birds feed by probing their beak in the mud, the functionality and evolutionary significance is not clear.

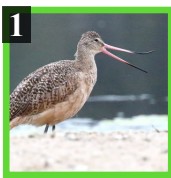 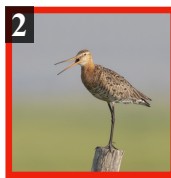 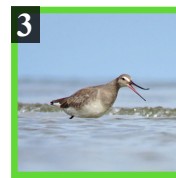 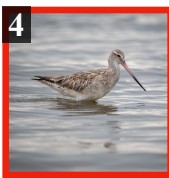 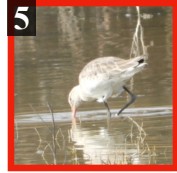

**Explanation**: (1) and (3) show pictures of godwits performing distal rhynchokinesis, as indicated by the upwards-bending upper beak. (2) shows a godwit with an open beak, but as the upper beak is straight, it is not performing distal rhynchokinesis. (4) shows a godwit with a straight, closed beak. (5) shows a godwit probing, so it not relevant as we can not infer if it is performing distal rhynchokinesis.

---

## 🔍 Strawberry poison-dart frog with the "la gruta" color morph from Isla Colon

Strawberry poison-dart frogs are known for their numerous color morphs, such as the common "blue jeans" morph with a red body with blue legs. Geographically isolated groups of frogs have extreme variability in coloration, but the reasons and mechanisms behind this are not clear, such as the importance of sexual selection of aposematic signaling. One such example is the "la gruta" color morph from Isla Colon, with a yellow-green base, possibly blue-ish legs, and dark dots.

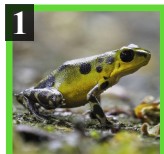 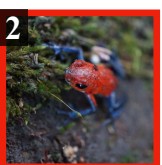 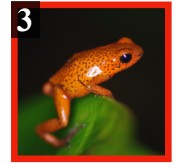 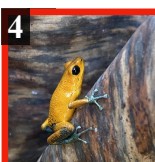 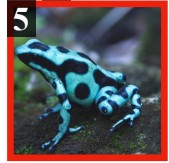 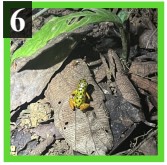

**Explanation**: (1) and (6) both show the "la gruta" morph with the characteristic yellow-green coloring and dark dots. (2) is the common "blue jeans" morph with a red both and blue legs. (3) and (4) are both different color morphs, while (5) is a Green and black poison dart frog, which is a different species.

---

## 🔍 Redwood trees with fire scars

Redwood trees like coast redwoods and giant sequoias area adapted to withstand fires, but forest mismanagement has lead to fires with unprecedented intensity. Pictures of their fire scars can help understand the impacts of wildfires on tree resilience, and fresh growth next to charred bark indicates that a tree has grown since the last fire. Fire also plays an important role in redwood reproduction, including opening up their cones and clearing the forest floor of competitive vegetation. Fire scars appear on redwood trees as blackened bark.

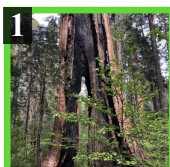 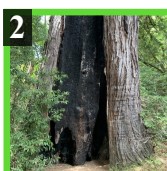 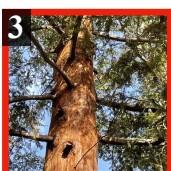 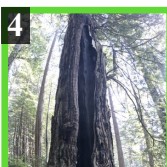 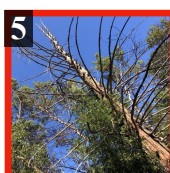

**Explanation**: All of these pictures show coast redwoods or giant sequoias. (1), (2) and (4) show fire scars, as evidenced by the blackened bark inside the trees. (3) Does not show evidence of fire scars, and (5) shows a tree which may have fallen over, but also does not show evidence of fire scars.

## B   Additional Details about INQUIRE

In Figure A1 we show histograms representing the number of labeled images and relevant images for each query from INQUIRE. We see that there is a long-tailed distribution for the number of relevant images per query, which ranges from 1 to 1150, with an average of 123 and median of 46 relevant images per query. In total, we labeled 149,022 images, of which 24,650 were relevant to their queries (or 24,336 unique images). As we use species filters and other steps to ensure our labeling is comprehensive (see Appendix H), we treat the rest of the iNat24 images as not relevant. This means that along with the existing image labels, we also have about 5 million weak negative labels per query, for a total of 1 billion weak labels.

In Table A1 we provide a breakdown of the number of queries of each of the four main types, including the average number of relevant images for each query. We note that this number varies widely. Species queries tend to have many relevant labels, while queries in categories like "Tracking and Identification" tend to have few relevant images.

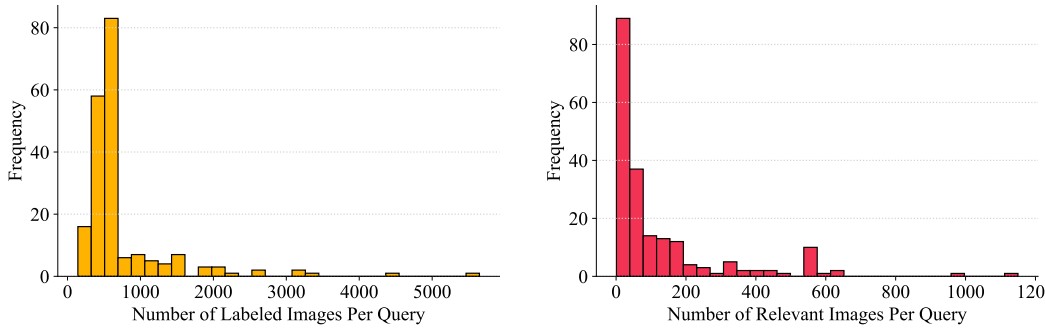

Figure A1: Collecting INQUIRE involved labeling a combined total of 149,022 candidate images across all 200 queries, yielding 24,650 relevant matches. These histograms show the number of images labeled per query, and the number of those that were relevant.

Table A1: INQUIRE queries can be grouped in to 16 categories. Here we provide ta list of these categories, the number of queries in each, and an example.

| Supercategory | Category | # Queries | Avg # Relevant | Example |
|---|---|---|---|---|
| Appearance | Health and Disease | 21 | 85 | black knot caused by a fungal pathogen |
| | Life Cycle and Development | 14 | 86 | juvenile bald eagle |
| | Sex identification | 14 | 136 | fiddler crab with an oversized chela |
| | Tracking and Identification | 13 | 27 | California Condor tagged with green 26 |
| | Unique appearances or morphs | 13 | 74 | albino american robin |
| Behavior | Cooperative and Social Behaviors | 18 | 25 | macaques engaging in mutual grooming behavior |
| | Defensive and Survival Behaviors | 11 | 47 | everted osmeterium |
| | Feeding and Hydration | 24 | 27 | Black Skimmer performing skimming |
| | Mating, Courtship, Reproduction | 12 | 32 | Alligator lizards mating |
| | Miscellaneous Behavior | 8 | 127 | spider monkey using its tail to hang on a branch |
| Context | Animal Structures and Habitats | 7 | 20 | a beaver dam across a stream |
| | Collected Specimens | 12 | 34 | measuring the body dimensions of a bee |
| | Human Impact | 17 | 46 | dehorned rhino |
| | Miscellaneous Context | 13 | 131 | Mushrooms growing in a fairy ring formation |
| | Parasitism and Symbiosis | 17 | 33 | Sharks with remoras attached |
| Species | Species ID | 25 | 419 | blue dragon nudibranch |

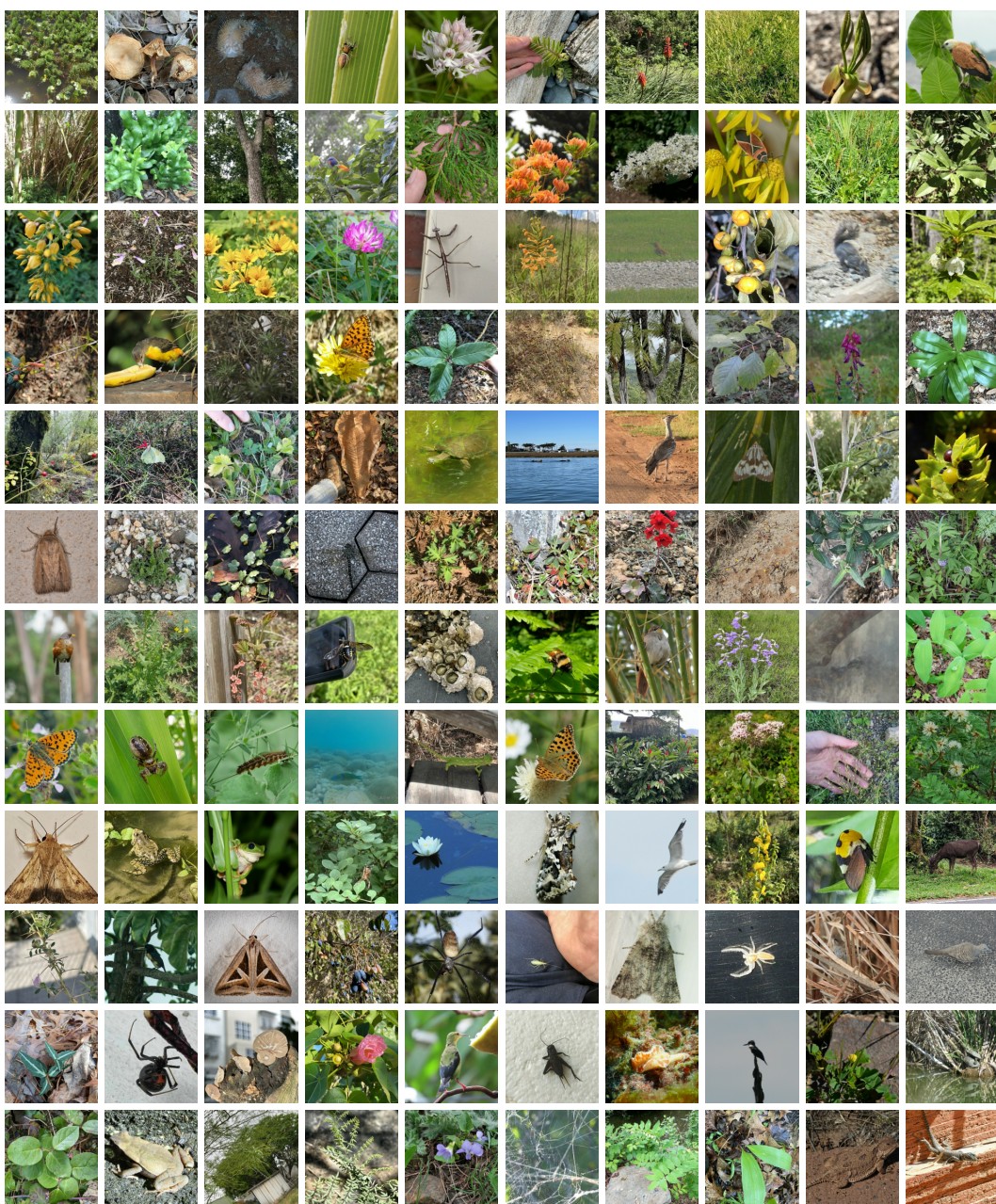

Figure A2: Random images from the iNat24 dataset. iNat24 contains five million images from 10,000 species classes.

## C Additional Details about iNat24

iNat24 contains 4,813,543 images for 9,959 species. Figure A2 shows examples of randomly chosen images from the dataset.

## D Geographic Range of INQUIRE and iNat24

In Figure A3 we show the geographic range of iNat24 observations and image from INQUIRE judged as relevant. We can see that the distribution of both is similar, which demonstrates that INQUIRE queries do not exhibit a strong geographic bias as compared to the iNat24 source data in the images

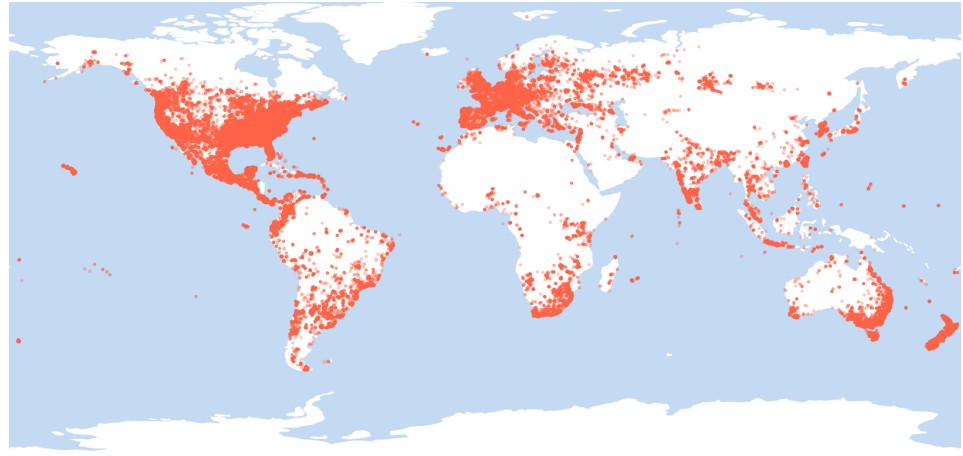

(a) Geographic distribution of all iNat24 images.

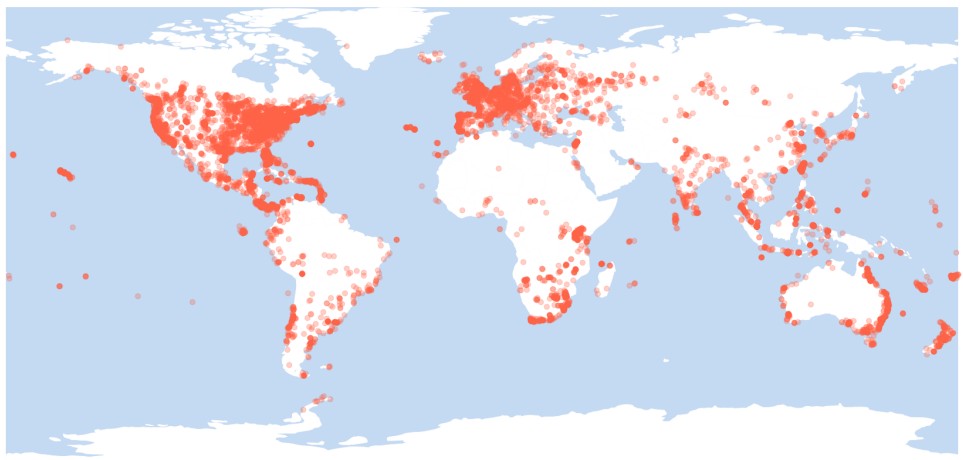

(b) Geographic distribution of the iNat24 images marked relevant for an INQUIRE query.

Figure A3: Here we compare the spatial distribution of the images in iNat24 to the relevant images in queries from INQUIRE. We can see that the distribution of both is similar. Both exhibit a bias towards North America, Europe, and parts of Australasia which is reflective of the spatial biases present in the iNaturalist platform.

that the queries correspond to. However, both exhibit a bias towards North America, Europe, and parts of Australasia which is reflective of the spatial biases present in the iNaturalist platform.

## E  Additional Results

In this section we provide detailed evaluation results for a range of additional CLIP models. We also break down results in detail by category, making it more clear what the strengths and weaknesses of each model are. These details results are shown in Table A2 for INQUIRE-FULLRANK and Table A3 for INQUIRE-RERANK.

## F  Computational Efficiency

**Embedding Generation.** The computational efficiency of a retrieval method is key to its real-world viability. In Figure A4 we estimate the computational cost for selected CLIP retrieval methods. Here,

Table A2: Detailed evaluation of INQUIRE-FULLRANK by category for a variety of embedding models. Results are reported in AP@50.

| Model | Sp. | Behavior | | | | | Context | | | | | Appearance | | | | |
|---|---|---|---|---|---|---|---|---|---|---|---|---|---|---|---|---|
| | Species ID | Miscellaneous Behavior | Defensive and Survival Behaviors | Cooperative and Social Behaviors | Mating, Courtship, Reproduction | Feeding and Hydration | Human Impact | Miscellaneous Context | Animal Structures and Habitats | Parasitism and Symbiosis | Collected Specimens | Tracking and Identification | Health and Disease | Unique appearances or morphs | Sex identification | Life Cycle and Development |
| bioclip | 21.1 | 2.4 | 0.0 | 0.4 | 0.1 | 0.2 | 0.0 | 9.9 | 0.0 | 0.8 | 0.1 | 0.6 | 0.7 | 0.4 | 7.1 | 2.4 |
| wildclip-t1t7-lwf | 13.1 | 14.5 | 4.7 | 4.5 | 6.7 | 7.2 | 10.2 | 5.6 | 8.6 | 3.5 | 2.8 | 5.2 | 8.9 | 14.5 | 2.9 | 0.8 |
| wildclip-t1 | 13.2 | 14.4 | 7.0 | 8.2 | 3.1 | 8.4 | 9.6 | 6.4 | 10.4 | 4.0 | 5.0 | 4.0 | 7.3 | 10.0 | 3.6 | 1.0 |
| rn50 | 13.8 | 17.0 | 5.8 | 6.3 | 5.4 | 6.2 | 6.5 | 14.2 | 6.3 | 4.9 | 9.2 | 1.9 | 7.2 | 15.4 | 2.9 | 1.0 |
| vit-b-32 | 16.1 | 15.7 | 5.4 | 7.1 | 7.0 | 6.2 | 9.9 | 11.7 | 6.3 | 6.1 | 10.9 | 4.1 | 4.8 | 12.7 | 6.5 | 1.1 |
| vit-b-16 | 19.0 | 17.0 | 9.6 | 11.7 | 8.8 | 8.4 | 15.2 | 9.0 | 13.4 | 5.8 | 14.2 | 9.1 | 9.6 | 23.3 | 6.6 | 0.8 |
| rn50x16 | 23.3 | 23.9 | 15.2 | 15.1 | 13.1 | 15.2 | 16.8 | 12.4 | 13.3 | 8.1 | 17.2 | 12.0 | 12.4 | 16.7 | 8.3 | 2.4 |
| vit-l-14 | 23.6 | 20.5 | 13.9 | 16.7 | 20.4 | 9.9 | 20.3 | 14.0 | 20.1 | 9.4 | 5.8 | 16.1 | 14.3 | 28.6 | 11.6 | 4.3 |
| vit-b-16-dfn | 28.3 | 26.5 | 14.0 | 12.7 | 15.3 | 19.1 | 19.3 | 16.7 | 21.5 | 9.6 | 12.8 | 10.5 | 9.4 | 20.4 | 14.8 | 5.1 |
| vit-l-14-dfn | 40.9 | 29.3 | 22.3 | 22.8 | 29.2 | 21.3 | 28.1 | 30.2 | 31.5 | 22.7 | 18.7 | 18.2 | 18.5 | 29.5 | 20.4 | 5.2 |
| siglip-vit-l-16-384 | 44.5 | 46.4 | 25.8 | 39.6 | 33.9 | 24.1 | 45.3 | 34.2 | 28.7 | 24.4 | 32.6 | 26.9 | **29.3** | 26.6 | 24.0 | 13.0 |
| siglip-so400m-14-384 | 42.6 | 48.5 | 29.0 | **42.5** | 29.8 | **30.0** | **45.8** | 34.0 | 32.9 | 24.8 | **43.0** | **42.2** | 29.2 | **38.4** | 25.6 | **18.0** |
| vit-h-14-378 | **52.7** | **51.0** | **35.5** | 41.2 | **44.4** | 29.8 | 45.7 | **42.5** | **38.0** | **26.1** | 27.2 | 30.2 | 27.6 | 26.0 | **28.4** | 15.7 |

Table A3: Detailed evaluation of INQUIRE-RERANK by category for a variety of embedding models. Results are reported in AP.

| Model | Sp. | Behavior | | | | | Context | | | | | Appearance | | | | |
|---|---|---|---|---|---|---|---|---|---|---|---|---|---|---|---|---|
| | Species ID | Miscellaneous Behavior | Defensive and Survival Behaviors | Cooperative and Social Behaviors | Mating, Courtship, Reproduction | Feeding and Hydration | Human Impact | Miscellaneous Context | Animal Structures and Habitats | Parasitism and Symbiosis | Collected Specimens | Tracking and Identification | Health and Disease | Unique appearances or morphs | Sex identification | Life Cycle and Development |
| bioclip | **41.1** | 45.4 | 20.7 | 26.7 | 33.7 | 21.8 | 27.5 | 43.2 | 16.1 | 29.6 | 39.4 | 20.4 | 26.1 | 21.5 | **39.7** | 32.5 |
| wildclip-t1t7-lwf | 37.4 | 55.7 | 36.2 | 31.4 | 31.0 | 21.2 | 47.8 | 38.9 | 18.5 | 26.0 | 37.5 | 23.8 | 30.8 | 34.6 | 26.6 | 29.8 |
| wildclip-t1 | 34.3 | 55.3 | 39.1 | 34.7 | 26.4 | 21.2 | 46.8 | 41.7 | 26.4 | 26.6 | 37.3 | 22.5 | 32.3 | 31.1 | 28.7 | 27.0 |
| rn50 | 35.2 | 55.7 | 25.2 | 31.8 | 29.8 | 24.2 | 36.4 | 45.0 | 23.2 | 29.4 | 42.5 | 19.5 | 32.1 | 37.9 | 27.4 | 28.4 |
| vit-b-32 | 37.0 | 53.9 | 26.3 | 32.7 | 28.3 | 24.8 | 43.2 | 44.8 | 22.2 | 31.5 | 41.5 | 21.4 | 28.6 | 34.2 | 25.3 | 25.9 |
| vit-b-16 | 37.1 | 58.8 | 33.2 | 35.9 | 30.7 | 24.5 | 46.4 | 42.0 | 23.5 | 27.6 | 43.3 | 31.4 | 31.1 | 42.0 | 24.0 | 25.9 |
| rn50x16 | 39.8 | 59.0 | 34.4 | 35.4 | 33.1 | 33.2 | 46.6 | 47.9 | 33.1 | 29.2 | 47.5 | 32.8 | 35.3 | 43.4 | 23.0 | 27.6 |
| vit-b-16-dfn | 31.5 | 53.0 | 32.7 | 33.3 | 36.0 | 33.4 | 47.0 | 40.0 | 24.1 | 29.8 | 39.4 | 29.6 | 28.0 | 36.3 | 23.2 | 29.6 |
| vit-l-14 | 37.6 | 55.0 | 34.4 | 41.8 | 38.6 | 30.6 | 53.8 | 48.7 | 36.2 | 28.4 | 37.5 | 36.3 | 37.1 | 49.8 | 21.5 | 29.5 |
| vit-l-14-dfn | 33.4 | 61.1 | 40.9 | 39.8 | 45.8 | 33.8 | 52.4 | 45.8 | 30.0 | 39.5 | 40.5 | 37.5 | 32.8 | 48.1 | 28.5 | 28.6 |
| siglip-vit-l-16-384 | 34.7 | **71.9** | 47.2 | 54.8 | **55.9** | 38.6 | 63.3 | 55.9 | 37.2 | **42.2** | 56.2 | 51.0 | **44.4** | 44.7 | 31.2 | 38.7 |
| siglip-so400m-14-384 | 38.8 | 70.7 | 49.0 | **56.6** | 51.1 | **43.2** | **66.1** | **56.1** | 37.2 | 40.4 | **62.3** | **66.7** | 44.0 | **65.3** | 30.2 | **49.6** |
| vit-h-14-378 | 28.6 | 70.0 | **55.1** | 52.0 | 53.8 | 39.8 | 60.8 | 44.1 | **37.4** | 40.2 | 45.3 | 51.6 | 39.6 | 41.2 | 25.3 | 32.0 |

the computational cost represents the total computational cost of generating all CLIP embeddings (the per-inference cost is provided by OpenCLIP [31]), and dividing by 200, the number of queries. However, we note that in practice, once all image embeddings are pre-computed and stored in an efficient nearest-neighbors index (e.g., Faiss [23]), each query takes milliseconds and thus its search cost is near-zero. The only significant computational cost will be that of performing inference on the query via the text encoder.

**Scaling Laws.** Figure A4-left also shows diminishing returns in AP@50 as the model size, and thus the computational cost, increases. When we plot the same data using a log-scaled x-axis in Figure A4-right, we observe a roughly linear trend between the log-scaled computational cost and the AP@50. While further study is required to fully characterize this particular trend, this result shows evidence of power law scaling similar to other machine learning tasks [16].

**Computational Resources Used.** All experiments we performed on A100 GPUs.

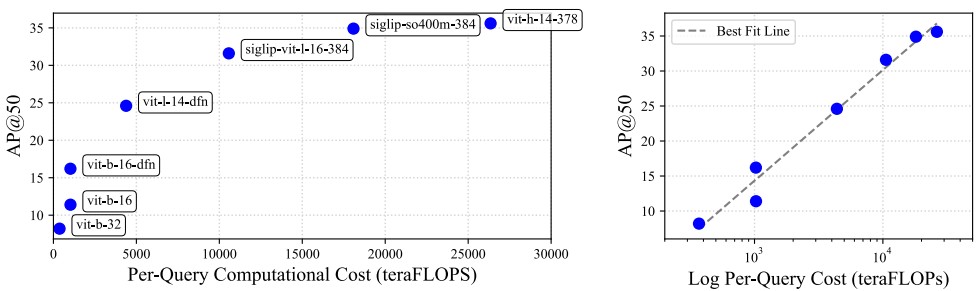

Figure A4: Computation cost of different CLIP models plotted against their performance on INQUIRE full dataset retrieval.

## G  Evaluation Metrics

Our primary evaluation metric is Average Precision at k (AP@k). We further report the Normalized Discounted Cumulative Gain (nDCG) and Mean Reciprocal Rank (MRR). While these metrics have been commonly used to evaluate text retrieval [71; 18], they have not found use in image retrieval due to the nonexistence of benchmarks like INQUIRE containing many relevant images for retrieval, rather than just one. Thus, we include them in our analysis to encourage their use in future image retrieval research. We note that the utilized AP@k metric uses a modified normalization factor suited to the retrieval setting. Existing image retrieval benchmarks typically evaluate using the recall@k metric (e.g., [40]), measuring if any of the top k images are relevant. While this makes sense in the setting where just one image is relevant, INQUIRE has potentially many relevant images and thus, we employ metrics that measure both relevance and ranking of retrievals. More detailed discussion about the metrics we used is provided below.

**Average Precision at k.**  Average Precision (AP) is a well-known metric computed by taking the weighted mean of precision scores at a set of thresholds. This metric has been adapted to the retrieval setting, where it possible to calculate the Average Precision at k (AP@k) among just the top k retrieved items. Since calculating AP@k requires both the relevance and position of the top k items, AP@k may be prefered over Precision at k (P@k) which does not use position. A number of AP@k variants have been proposed [10; 27; 71], taking the general form:

$$AP@k = \frac{\sum_{i=1}^{k} P@i \cdot rel(i)}{NF} \tag{1}$$

where $Pr@i$ is the precision at $i$ (i.e., among the first $i$ items), $rel(i) \in \{0, 1\}$ is the binary relevance score, and $NF$ is a normalization factor.

In a typical implemenation of AP we would see $NF = r$, the total number of relevant items in the top $k$. However in a retrieval setting with a total of $R$ relevant items, this normalization technique creates a problematic and unintuitive situation where promoting an item into the top k retrievals can decrease the score.

In particular, consider the situation where we have 100 images of which 2 are relevant and 98 are not relevant. Using a normalization factor of $NF = R$, we measure AP@5 for the following two top-5 retrievals:

1. Ordered retrieval relevance: $(1, 0, 0, 0, 0) \implies$ AP@5 = 1
2. Ordered retrieval relevance: $(1, 0, 0, 0, 1) \implies$ AP@5 = 0.7

We observe that promoting a relevant item into the top 5 resulted in a decreased AP@5, which is undesirable. Our criteria for an AP@ metric is that (1) the measure strictly increases whenever a relevant document is promoted into the top-k, and (2) the has a full range of 0 to 1. Of the range

of proposed AP@k variants [10; 27; 71], just [71] meets our desired criteria This modified average precision normalizes using $min(k, R)$. In the case above, we now have $NF = min(k, R) = min(5, 2) = 2$, yielding:

1. Ordered retrieval relevance: $(1, 0, 0, 0, 0) \implies$ AP@5 = 0.5

2. Ordered retrieval relevance: $(1, 0, 0, 0, 1) \implies$ AP@5 = 0.7

Our end-to-end retrieval evaluations use AP@k with this desirable normalization factor of $NF = min(k, R)$. Since the reranking challenge evaluates solely using the fixed set, the normalization factor for this challenge is always $r$, the number of relevant items within the top k.

For further discussion of Average Precision at k, we refer readers to [17].

**nDCG**. Normalized discounted cumulative gain is a weighted ranking metric that considers the relative ordering of the retrieved items. To compute nDCG@k for a single query, first we compute the discounted cumulative gain at k (DCG@K):

$$DCG_k = \sum_{i=1}^{k} \frac{rel(i)}{\log_2(i + 1)} \tag{2}$$

where $rel(i) \in \{0, 1\}$ is the binary relevance score for the $i$th retrieved item. Then, we define the ideal DCG at k (IDCG@k) as the maximum achievable DCG@k:

$$IDCG_k = \sum_{i=1}^{\min(k,R)} \frac{rel(i)}{\log_2(i + 1)} \tag{3}$$

where $R$ is the total number of relevant items for the query. Finally, we can compute $nDCG@k$ as

$$nDCG_k = \frac{DCG_k}{IDCG_k} \tag{4}$$

where the normalization by $IDCG@k$ allows $nDCG_k$ to range fully between the interval 0 to 1.

**MRR**. Mean reciprocal rank is a measure for the rank of the first correct retrieval. MRR can be computed as

$$MRR = \frac{1}{Q} \sum_{i=1}^{Q} \frac{1}{rank(i)} \tag{5}$$

where $Q$ is the number of queries, and $rank(i)$ gives the rank of the first relevant retrieval for the $i$th query (1 for 1st position, 2 for 2nd position, etc.). If no relevant retrievals are present in the retrieved list, we let $rank(i) = \infty$, i.e., $1/rank(i) = 0$.

# H  iNat24 Image Collection and INQUIRE Annotation Protocol

In this sections we describe in detail our data collection protocol for collecting the iNat24 dataset and annotating the INQUIRE benchmark.

## H.1  iNat24 Dataset Curation

We follow a similar paradigm used to organize the iNaturalist Competition Datasets from 2017 [69], 2018 [1], 2019 [1], and 2021 [70]. For the 2024 version we start from an iNaturalist observation database export generated on 2023-12-30. Observations are then filtered to include only those observed in the years 2021, 2022, or 2023. This ensures the images in iNat24 are unique and do not overlap with images from prior dataset versions (e.g., iNat21 [70] only contains images up until September 2020). To utilize the iNat21 taxonomy (for easy compatibility with that dataset) we detect taxonomic changes between the iNat21 taxonomy and the iNaturalist taxonomy included in the 2023-12-30 database export. We then modify species labels (where necessary) so that observations

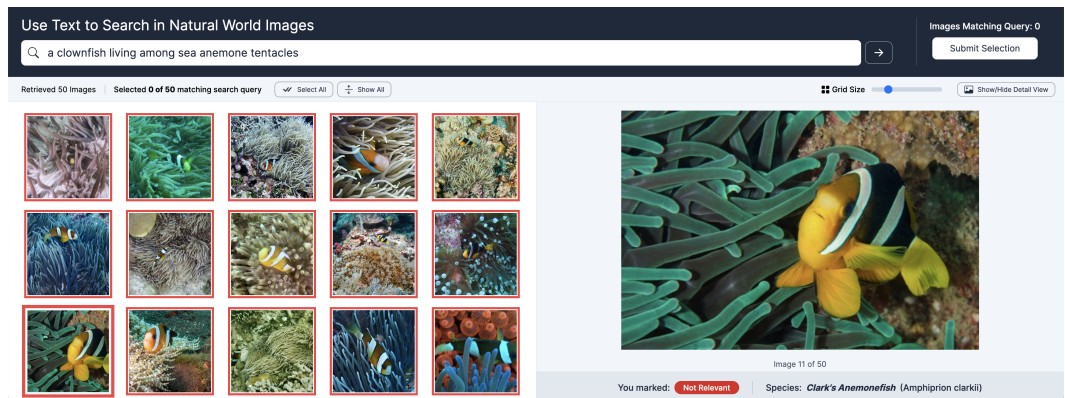

Figure A5: Here is display a screen shot of the online annotation tool we developed for annotation. The tool supports CLIP similarity search and species filtering.

conform to the iNat21 taxonomy. Some of these taxonomic changes can be quite complicated (splits, merges, etc.) resulting in cases where an iNat21 species is no longer valid, however we are able to recover 9,959 out of the original 10,000 species from iNat21. We then filter to include observations exclusively from the iNat21 taxonomy. Additional filtering ensures that all observations have valid metadata (i.e., location and time information) and that associated image files are not corrupted. These steps result in a candidate set of 33M observations to sample from to build the iNat24 dataset.

Our process of selecting the set of images to include for each species in the iNat24 dataset deviates from the prior dataset building schemes [69; 70]. Random sampling of observations, or even random sampling from unique users, generates collections of images that are biased towards North America and Europe. To decrease this bias we sample from spatio-temporal clusters of "observations groups". Observation groups are formed by grouping observations together if they are observed on the same day within 10km of each other, regardless of the observer. When sampling observations for a species, we cluster their associated observation groups using a spatio-temporal distance metric and then sample one observation per cluster in a round-robin fashion until we hit a desired sample size. When sampling within a cluster, we prioritize novel observation groups and novel users. We sample at most 550 observations per species to include in iNat24. This sampling process results in a total of 4,816,146 images for 9,959 species.

Unlike previous versions of the iNaturalist dataset, we performed one final round of filtering to remove images that are inappropriate for a research dataset or not relevant for the query. We use the INQUIRE annotation process to find images containing human faces, personally identifiable information, "empty" images, images of spectrograms, etc.. We additionally run a RetinaFace [21] Resnet50 face detection model across the entire dataset, and manually inspect all high confidence predictions. In total this filtered out an additional 2,603 images. The final dataset contains 4,813,543 images for 9,959 species.

The iNat24 dataset does not have a validation or test split, i.e., all observations are assigned to the train split. The validation and test splits can be used from the iNat21 dataset to benchmark classification performance. As in previous years, we keep only the primary image for each observation, and resize all images to have a max dimension of 500px on the longest side. All images have three channels and are stored as jpegs. We provide location, time, attribution, and licensing information in the associated json file.

## H.2  Data Annotation

Image annotation was performed by a carefully selected team of paid MSc students or equivalent, many with expertise in ecology allowing for labeling of difficult queries. Annotators were instructed to label all candidate images as either relevant (i.e., positive match) or not relevant (i.e., negative

match) to the query, and to mark an image as not relevant if there was reasonable doubt as to its relevance. At this stage, queries that were deemed very easy, not comprehensively labeled, or otherwise not possible to label were excluded from the benchmark.

The annotation itself is performed using a custom interface that we developed that shows the top retrievals given a text query and optionally allows the user to filter based on the species label. A screen shot of the tool is displayed in Figure A5. The retrievals are ordered by CLIP ViT-H/14 [24] similarity to the query text. Annotators generally label at least 500 images per query.

We comprehensively labeled the dataset primarily through the use of species filters for a single or a group of species. For example, to thoroughly label the query *"Black Skimmer performing skimming"*, a single species filter (Black Skimmer) was utilized while for the query *"flamingo standing on one leg"*, four different species filters were needed to account for all the flamingo species included in iNat24 (Lesser Flamingo, Chilean Flamingo, Greater Flamingo, and American Flamingo). Using species filters in this way allows us to sufficiently reduce the search space for these queries to comprehensively label iNat24 for all possible matches.

When a query corresponds to a very large number of species, or no species in particular (e.g., *"an image containing a photographic reference scale with a color swatch"*), we label using just the top CLIP retrievals without any filters. In this case, we tend to label a significantly larger number of images, and we label until at least 100 images in a row are negative indicating that the set of positives has been exhausted. If this condition is not met after at a large number of labels, or the annotator otherwise believes that comprehensive labeling is not possible, we do not use the query. We note that the quality of our comprehensive labeling in this case is limited by the CLIP model's ability to surface relevant positives, so any missed positives with lower relevance score could be left unlabeled. This affects only 11 of our 200 queries for which we label without species filters. This is a primary motivator behind the large number of images labeled per query (i.e., >500). However, if there were indeed missed positives, then we would expect the CLIP ViT-H/14 model used for labeling to perform unexpectedly well, as higher quality models that surface missed positive image would be penalized as these would be considered negative at evaluation time. Yet, our evaluation in Table 2 shows that SigLIP, which on OpenCLIP's [31] retrieval evaluation performs only marginally better than CLIP ViT-H/14, achieves a comparable score. This result suggests that our dataset does not suffer from a significant missed positive issue.

Creating INQUIRE involved labeling 149,022 images, of which 24,650 were relevant to their queries. Labeling took place over a total of about 113 hours, so the average time spent labeling is 34 minutes per query or 2.7 seconds per image.

## H.3  Data Format and Structure

**iNat24**. iNat24 is provided as a metadata file and a tar file containing all images. The metadata file is given in the commonly used JSON COCO format. The information in this metadata file includes each image's ID, file path, width, height, image license, rights holder, taxonomic classification, latitude, longitude, location uncertainty, and date.

**INQUIRE**. The INQUIRE benchmark is provided as a two CSV files. The first is a list of queries, where each row includes fields for the query id, query text, organism category, query category type, and query category. The second file is a list of annotations, where row corresponds includes fields for the query id, image id, and relevance label. The image id can be matched to the iNat24 metadata to get additional information mentioned above, such as the taxonomy, date, and geographic location.

## H.4  Ethical Considerations

**Copyright and Licensing**. We adhere strictly to copyright and licensing regulations. All images included in the dataset fall under a license allowing copying and redistribution. In particular, all images are licensed under one of the following: CC BY 4.0, CC BY-NC 4.0, CC BY-NC-ND 4.0, CC BY-NC-SA 4.0, CC0 1.0, CC BY-ND 4.0, or CC BY-SA 4.0.

**Data Privacy and Safety**. Although users approved all images considered for research use, we take further steps to ensure data privacy and safety. We filter all images for content that is contains personally identifiable information or images of people. We do not exclude most images containing gore, as these are often ecologically relevant, e.g., using image of road-killed animals to asses impacts of roads on biodiversity.

**Violations of Rights**. We respect the rights of iNaturalist community volunteer observers by constructing iNat2024 using only images and metadata appropriately licensed by their respective creators for copying, distribution, and non-commercial research use. Nevertheless, we bear responsibility in case of a violation of rights.

**Participant Risks**. We received internal ethical approval for our query collection and data labeling (Edinburgh Informatics Ethics Review Panel 951781 and MIT Committee on the Use of Humans as Experimental Subjects Protocol 2404001276).

### H.5 Participant Compensation

We hired annotators at the equivalent of $15.50 per hour and spent a total of $1750 on annotation.

### H.6 Annotation Instructions

The instructions provided to annotators are included below.

# Annotation Guide

March 25 2024

## Introduction

- Your goal is to label if an image matches a search query
- Images matching a query are called "relevant"
- You should make sure to label **all** the relevant images

*"Dog by a river"*

(example query)

✅ **Relevant**

❌ **Not Relevant**

---

This is the annotation page

---

Enter a **query** here, then press enter (or the arrow button) to search.

---

You can also add advanced **filters**. Press "Advanced filters" to open additional options (specifically, species filtering)

---

After press "Advanced filters" you'll see this extra field. Select a species by its common or scientific name

---

As you type, some suggestions will appear

---

You'll know a species filter is on when there is a green checkmark and the species name is green

## When you search, you'll see the first 50 results

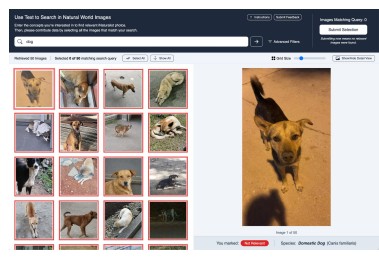

**The grid shows all the images**

**The detail view shows information about the selected image**

## The grid view shows all images and their labels

By default, all images are marked "Not relevant". You can toggle the label by clicking on it.

Relevant images will have a **green** border, while Not Relevant images will have a **red** border

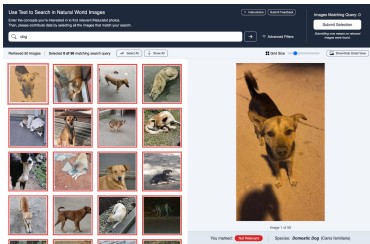

## The detail view shows useful information about one image

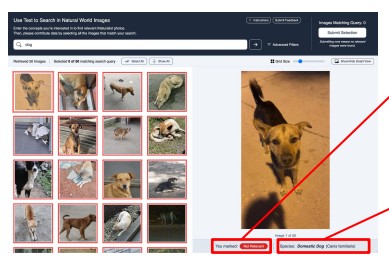

You see the image's label, which is either "**Relevant**" or "**Not Relevant**". All images start out Not Relevant

The species name is useful when the query is relevant for only some species

## Keyboard shortcuts can help make labeling much faster!

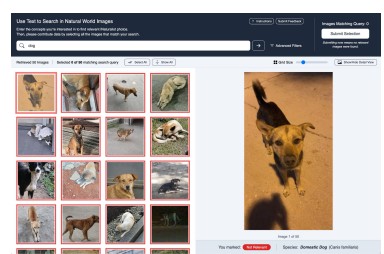

**LEFT**: Show the previous image

**RIGHT**: Show the next image

**ENTER or SPACE**: Toggle the label on the selected image

## When you finish labeling all the images, press "Submit"

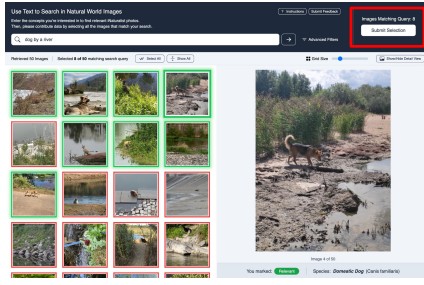

## Mark that you completed labeling a query in the sheet

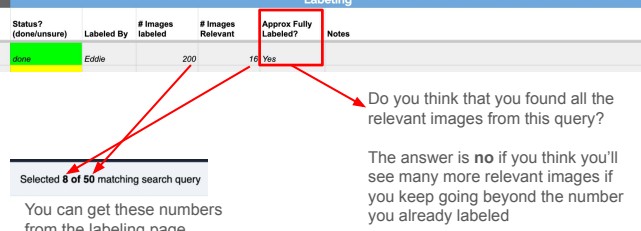

Do you think that you found all the relevant images from this query?

The answer is **no** if you think you'll see many more relevant images if you keep going beyond the number you already labeled

You can get these numbers from the labeling page

## "Rules of thumb"

- Aim to label at least **500 images per task** (a task can have multiple queries)
- If a task has many queries, you can label fewer images per query (e.g. 200 images per query)
- The amount you label depends on how many relevant images you see. e.g. if you see barely any relevant images by the end of the currently open 200 images, you don't need to keep going

## I  Multi-Modal Model Prompting Details

We include the various prompts used in our evaluation of large multimodal models in Table A4.
We note that while we aim to keep the prompt broadly the same across models, they are ultimately
different due to different prompting requirements for each model.

Table A4: Format of the text prompts used by the large multimodal models.

| | |
|---|---|
| BLIP-2 | `Does this picture show {query}?\nAnswer the question with either "Yes" or "No" and nothing else.` |
| InstructBLIP | `Does this picture show {query}?\nAnswer the question with either "Yes" or "No" and nothing else.` |
| LLaVA-v1.5 | `USER: <image>\nDoes this picture show {query}?  Answer the question with either "Yes" or "No" and nothing else.\nASSISTANT:` |
| LLaVA-v1.6-7b | `[INST] <image>\nDoes this picture show {query}? Answer the question with either "Yes" or "No" and nothing else.  [/INST]` |
| LLaVA-v1.6-34b | `<|im_start|>system\nAnswer the questions.<|im_end|> <|im_start|>user \n<image>\nDoes this image show "{query}"?  Answer the question with either "Yes" or "No".  <|im_end|><|im_start|>answer\n` |
| PaliGemma | `Q: Does this picture show {query}?  Respond with yes or no.\nA:` |
| VILA-v1.5-13B | `<image>\n Does this picture show {query}?\nAnswer the question with either "Yes" or "No" and nothing else.` |
| VILA-v1.5-40B | `<image>\n Does this picture show {query}?\nAnswer the question with either "Yes" or "No" and nothing else.` |
| GPT-4V | `Does this picture show exactly "{query}"?\nAnswer the question with either "Yes" or "No" and nothing else.` |
| GPT-4o | `Does this picture show exactly "{query}"?\nAnswer the question with either "Yes" or "No" and nothing else.` |

## J  Full List of INQUIRE Queries

Table A5 lists all INQUIRE queries.

Table A5: INQUIRE includes 200 queries across a range of categories. This table lists all 200 queries
along with the supercategory and category that they belong to.

| Query | Supercategory | Category |
|---|---|---|
| Dead hog-nosed skunk | Appearance | Health and Disease |
| sick cassava plant | Appearance | Health and Disease |
| black knot caused by a fungal pathogen | Appearance | Health and Disease |
| beached orca | Appearance | Health and Disease |
| common murre beached carcass | Appearance | Health and Disease |
| southern aligator lizard with cut tail | Appearance | Health and Disease |
| moose with hair loss | Appearance | Health and Disease |
| elk with hair loss | Appearance | Health and Disease |
| red fox showing signs of sarcoptic mange | Appearance | Health and Disease |
| fire pink with dark-colored anthers | Appearance | Health and Disease |
| common lilac with powdery mildew | Appearance | Health and Disease |

| | | |
|---|---|---|
| Redwood tree with fire scars | Appearance | Health and Disease |
| Wolf spider with limb loss | Appearance | Health and Disease |
| Moorish Gecko with regenerated tail | Appearance | Health and Disease |
| an immature bald eagle | Appearance | Life Cycle and Development |
| Swallowtail butterfly caterpillar camouflaged as bird droppings | Appearance | Life Cycle and Development |
| A cicada in the process of shedding its exoskeleton | Appearance | Life Cycle and Development |
| monkey slug caterpillar | Appearance | Life Cycle and Development |
| laysan albatross mostly in dark mottled brown plumage | Appearance | Life Cycle and Development |
| penguin during molting period | Appearance | Life Cycle and Development |
| Lilac Bonnet with edges turnt up revealing its gills | Appearance | Life Cycle and Development |
| Octopus Stinkhorn fungus emerging from casing | Appearance | Life Cycle and Development |
| breeding adult Dunlin | Appearance | Life Cycle and Development |
| Cooper's Hawk in adult plumage | Appearance | Life Cycle and Development |
| breeding adult Black-bellied Plover | Appearance | Life Cycle and Development |
| Cart-Rut Shell snail egg mass | Appearance | Life Cycle and Development |
| A female pheasant | Appearance | Sex identification |
| fiddler crab with an oversized chela | Appearance | Sex identification |
| Male crimsonband wrasse | Appearance | Sex identification |
| Eurasian Black Grouse male | Appearance | Sex identification |
| Rusty tussock moth adult female | Appearance | Sex identification |
| male velvet ant | Appearance | Sex identification |
| Male Xanthagrion erythroneurum | Appearance | Sex identification |
| female or immature evening grosbeak | Appearance | Sex identification |
| male common green darner | Appearance | Sex identification |
| male ruby-throated hummingbird in flight | Appearance | Sex identification |
| female beautiful demoiselle | Appearance | Sex identification |
| male Northern Elephant Seal | Appearance | Sex identification |
| adult male Misumena vatia | Appearance | Sex identification |
| Tagged swan | Appearance | Tracking and Identification |
| a north island robin tagged with colored leg bands | Appearance | Tracking and Identification |
| California Condor tagged with green 26 | Appearance | Tracking and Identification |
| a lion with a collar around its neck | Appearance | Tracking and Identification |
| elephant with radio collar | Appearance | Tracking and Identification |
| Cheetah with radio collar | Appearance | Tracking and Identification |
| Rhino with ear notches | Appearance | Tracking and Identification |
| an image showing a humpback whale fluke with clearly identifiable markings | Appearance | Tracking and Identification |
| A grey-tailed tattler with leg bands | Appearance | Tracking and Identification |
| Black Skimmer with a leg band | Appearance | Tracking and Identification |
| a bighorn sheep with a tracking collar around its neck | Appearance | Tracking and Identification |
| Tortoise with a radio tag on its shell | Appearance | Tracking and Identification |
| a male mandarin duck in breeding plumage | Appearance | Unique appearances or morphs |
| Strawberry poison-dart frog with the "la gruta" color morph from Isla Colon | Appearance | Unique appearances or morphs |
| Melanistic leopard | Appearance | Unique appearances or morphs |
| Melanistic jaguar | Appearance | Unique appearances or morphs |
| a peach-faced Lovebird with the turquoise mutation | Appearance | Unique appearances or morphs |
| albino american robin | Appearance | Unique appearances or morphs |
| A stoat with mainly white fur | Appearance | Unique appearances or morphs |

| fire salamander with a barred color pattern | Appearance | Unique appearances or morphs |
|---|---|---|
| brown-colored black bear | Appearance | Unique appearances or morphs |
| eastern gray squirrel displaying melanistic pelage | Appearance | Unique appearances or morphs |
| Fly Agaric in yellow form | Appearance | Unique appearances or morphs |
| Axanthism in a green frog (Lithobates clamitans) | Appearance | Unique appearances or morphs |
| A meadowlark vocalizing | Behavior | Cooperative and Social Behaviors |
| A close-up of an ant carrying a leaf | Behavior | Cooperative and Social Behaviors |
| cheetah with cubs | Behavior | Cooperative and Social Behaviors |
| grebe with babies on its back | Behavior | Cooperative and Social Behaviors |
| macaques engaging in mutual grooming behavior | Behavior | Cooperative and Social Behaviors |
| A tamandua anteater pup clinging to its mother's back | Behavior | Cooperative and Social Behaviors |
| Emergence of large colony of mexican free-tailed bats | Behavior | Cooperative and Social Behaviors |
| a herd of more than 10 impalas | Behavior | Cooperative and Social Behaviors |
| canada geese flying in v-formation | Behavior | Cooperative and Social Behaviors |
| mountain goat climbing rocky outcrops with its young | Behavior | Cooperative and Social Behaviors |
| jungle babblers allopreening | Behavior | Cooperative and Social Behaviors |
| a sandhill crane couple with their chicks | Behavior | Cooperative and Social Behaviors |
| couple of black-bellied whistling ducks with their youngs sharing parenting duties | Behavior | Cooperative and Social Behaviors |
| Eurasian Blackbird vocalizing | Behavior | Cooperative and Social Behaviors |
| Sage Thrasher vocalizing | Behavior | Cooperative and Social Behaviors |
| male Red-winged Blackbird vocalizing | Behavior | Cooperative and Social Behaviors |
| picture showing more than fifty Velvety Tree Ants | Behavior | Cooperative and Social Behaviors |
| picture showing more than fifty Mediterranean Acrobat Ants | Behavior | Cooperative and Social Behaviors |
| Wolf spider carrying spiderlings on its back | Behavior | Cooperative and Social Behaviors |
| Scorpion with young on its back | Behavior | Cooperative and Social Behaviors |
| Two giraffes | Behavior | Cooperative and Social Behaviors |
| A mongoose standing upright alert | Behavior | Defensive and Survival Behaviors |
| Gazelle being vigilant/looking around | Behavior | Defensive and Survival Behaviors |
| everted osmeterium | Behavior | Defensive and Survival Behaviors |
| vigilant prairie dog stands guard | Behavior | Defensive and Survival Behaviors |
| Inflated pufferfish | Behavior | Defensive and Survival Behaviors |
| killdeer feigning injury | Behavior | Defensive and Survival Behaviors |
| white-tailed deer flagging its tail | Behavior | Defensive and Survival Behaviors |
| white-tailed deer lifting forefoot for foot-stomp | Behavior | Defensive and Survival Behaviors |
| moray eel with open mouth poking head out of burrows or crevices | Behavior | Defensive and Survival Behaviors |
| puffins carrying food | Behavior | Feeding and Hydration |
| black-winged kite carrying prey in its talons | Behavior | Feeding and Hydration |
| parrotfish feeding | Behavior | Feeding and Hydration |
| Elephants at a watering hole | Behavior | Feeding and Hydration |
| A male and female cardinal sharing food | Behavior | Feeding and Hydration |
| hyenas eating a kill | Behavior | Feeding and Hydration |
| A godwit performing distal rhynchokinesis | Behavior | Feeding and Hydration |
| Black Skimmer performing skimming | Behavior | Feeding and Hydration |
| fruit bat eating fruit upside down | Behavior | Feeding and Hydration |
| Honey Bee carrying pollen baskets | Behavior | Feeding and Hydration |
| macaque breastfeeding its young | Behavior | Feeding and Hydration |
| great golden digger wasp carrying an orthopteron | Behavior | Feeding and Hydration |

| | | |
|---|---|---|
| red-tailed hawk perched on a utility pole | Behavior | Feeding and Hydration |
| blue jay eating whole peanuts | Behavior | Feeding and Hydration |
| water snake feeding on fish | Behavior | Feeding and Hydration |
| Yellow-faced Honeyeater in birdbath | Behavior | Feeding and Hydration |
| Northern Mockingbird carrying out its food | Behavior | Feeding and Hydration |
| Leafhopper Assassin Bug predating lady beetle | Behavior | Feeding and Hydration |
| Milkweed Assassin Bug predating a bee or wasp | Behavior | Feeding and Hydration |
| Surgeonfish grazing on algae | Behavior | Feeding and Hydration |
| Butterflyfish feeding on brain coral | Behavior | Feeding and Hydration |
| A peacock male displaying its feathers | Behavior | Mating, Courtship, Reproduction |
| A glowworm exhibiting bioluminescence | Behavior | Mating, Courtship, Reproduction |
| A male frigatebird with an inflated throat pouch | Behavior | Mating, Courtship, Reproduction |
| Elk bugling during the rut | Behavior | Mating, Courtship, Reproduction |
| baboon with swollen red bottom | Behavior | Mating, Courtship, Reproduction |
| pair of great crested grebes potentially performing the weed dance | Behavior | Mating, Courtship, Reproduction |
| male smooth newt with developed crest | Behavior | Mating, Courtship, Reproduction |
| firebugs mating | Behavior | Mating, Courtship, Reproduction |
| Hübner's Wasp Moth mating | Behavior | Mating, Courtship, Reproduction |
| Alligator lizards mating | Behavior | Mating, Courtship, Reproduction |
| Water Frogs in amplexus position | Behavior | Mating, Courtship, Reproduction |
| Dolphins performing acrobatics | Behavior | Miscellaneous Behavior |
| elephant covered in mud or dirt | Behavior | Miscellaneous Behavior |
| spider monkey using its tail to hang on a branch | Behavior | Miscellaneous Behavior |
| flamingo standing on one leg | Behavior | Miscellaneous Behavior |
| African Buffalo wallowing in mud | Behavior | Miscellaneous Behavior |
| Eastern Red Bat in flight | Behavior | Miscellaneous Behavior |
| Big Brown Bat roosting | Behavior | Miscellaneous Behavior |
| Young sea turtles heading towards the ocean | Behavior | Miscellaneous Behavior |
| A Gila Woodpecker inside its nest in a Saguaro cactus cavity | Context | Animal Structures and Habitats |
| A satin bowerbird's bower ornamented with blue objects | Context | Animal Structures and Habitats |
| potter wasp nest | Context | Animal Structures and Habitats |
| Hamerkop collecting nesting material | Context | Animal Structures and Habitats |
| male purple finch on a nest box | Context | Animal Structures and Habitats |
| a beaver dam across a stream | Context | Animal Structures and Habitats |
| measuring the body dimensions of a bee | Context | Collected Specimens |
| camera trap photo of a cougar captured in the day-time | Context | Collected Specimens |
| camera trap photo of a bobcat captured in the nigh-time | Context | Collected Specimens |
| camera trap photo of a stag red deer | Context | Collected Specimens |
| camera trap photo of chital with its head down | Context | Collected Specimens |
| camera trap photo of a springbok drinking water | Context | Collected Specimens |
| leopard on a road | Context | Human Impact |
| a hermit crab using plastic waste as a shell | Context | Human Impact |
| Fishing net on a reef | Context | Human Impact |
| a possum on a power line | Context | Human Impact |
| elephant near a fence | Context | Human Impact |
| birds with wind turbines | Context | Human Impact |
| bird dead in front of a window | Context | Human Impact |
| bird caught in a net | Context | Human Impact |

| | | |
|---|---|---|
| dehorned rhino | Context | Human Impact |
| brown bear near vehicle | Context | Human Impact |
| human handling a bat with bare hands | Context | Human Impact |
| raccoon observed in urban setting | Context | Human Impact |
| Maple tree with signs of tapping | Context | Human Impact |
| Mushrooms growing in a fairy ring formation | Context | Miscellaneous Context |
| an image containing a photographic reference scale with a color swatch | Context | Miscellaneous Context |
| a microscopy slide showing the cellular structure of a plant | Context | Miscellaneous Context |
| dorsal side of mourning cloak butterfly | Context | Miscellaneous Context |
| ventral side of mourning cloak butterfly | Context | Miscellaneous Context |
| Indigo Milk Cap underneath view | Context | Miscellaneous Context |
| hot lips plant with blue fruits | Context | Miscellaneous Context |
| great blue heron with visible water reflection | Context | Miscellaneous Context |
| Scarlet Waxy Cap with visible gills | Context | Miscellaneous Context |
| an oxpecker on a zebra | Context | Parasitism and Symbiosis |
| bird perched on a hippo | Context | Parasitism and Symbiosis |
| Sea turtle with algae on its shell | Context | Parasitism and Symbiosis |
| Zebra and wildebeest grazing together | Context | Parasitism and Symbiosis |
| Sharks with remoras attached | Context | Parasitism and Symbiosis |
| bananaquit pollinating flower | Context | Parasitism and Symbiosis |
| lorikeet pollinating flower | Context | Parasitism and Symbiosis |
| a nest with eggs displaying brood parasitism by a cowbird | Context | Parasitism and Symbiosis |
| giant resin bee feeding on sunflower | Context | Parasitism and Symbiosis |
| Cat's Eye Snail covered by green algae | Context | Parasitism and Symbiosis |
| Channeled Applesnail covered by green algae | Context | Parasitism and Symbiosis |
| Chinese Mystery Snail covered by green algae | Context | Parasitism and Symbiosis |
| cross orbweaver | Species | Species ID |
| Carpobrotus ice plant | Species | Species ID |
| Zebra Mussel | Species | Species ID |
| Spotted Lanternfly | Species | Species ID |
| bridal veil stinkhorn mushroom | Species | Species ID |
| Green Shore Crab | Species | Species ID |
| Parasitic Honey Mushrooms | Species | Species ID |
| Green and black poison dart frog | Species | Species ID |
| Chain Tunicate | Species | Species ID |
| blue dragon nudibranch | Species | Species ID |
| close-up of shagbark hickory tree bark | Species | Species ID |
| close-up of sweet cherry tree bark | Species | Species ID |
| close-up of sugar maple leaf | Species | Species ID |
| close-up of silver maple leaf | Species | Species ID |
| Death cap mushroom | Species | Species ID |
| Canada goldenrod (Solidago canadensis) | Species | Species ID |
| Japanese knotweed | Species | Species ID |
| kahili ginger plant with open fruit capsules show-ing seeds | Species | Species ID |
| a rosy wolfsnail | Species | Species ID |
| Purple Sea Urchin | Species | Species ID |
| Sunflower Sea Star | Species | Species ID |

## K  Datasheet

### K.1  Motivation

**For what purpose was the dataset created?**  Was there a specific task in mind? Was there a specific gap that needed to be filled? Please provide a description.

- The purpose of INQUIRE is to provide a challenging benchmark for text-to-image retrieval on natural world images. Prior retrieval datasets are small and do not possess a challenge for existing models, with many being adaptations of captioning datasets. These datasets also have exactly one positive match for each query, which differs significantly from real-world retrieval scenarios where many images can be matches. The initial release of INQUIRE includes 200 queries comprehensively labeled over a pool of five million natural world images. For more information see Section 3.

**Who created this dataset (e.g., which team, research group) and on behalf of which entity (e.g., company, institution, organization)?**

- INQUIRE and iNat24 were created by a group of researchers from the following affiliations: iNaturalist, the Massachusetts Institute of Technology, University College London, University of Edinburgh, and University of Massachusetts Amherst. The dataset was created from data made publicly available by the citizen science platform iNaturalist [2].

**What support was needed to make this dataset?** (e.g.who funded the creation of the dataset? If there is an associated grant, provide the name of the grantor and the grant name and number, or if it was supported by a company or government agency, give those details.)

- Funding for annotation was provided by the Generative AI Laboratory (GAIL) at the University of Edinburgh. In addition, team members were supported in part by the Global Center on AI and Biodiversity Change (NSF OISE-2330423 and NSERC 585136) and the Biome Health Project funded by WWF-UK.

**Any other comments?**

- N/A

### K.2  Composition

**What do the instances that comprise the dataset represent (e.g., documents, photos, people, countries)?**  Are there multiple types of instances (e.g., movies, users, and ratings; people and interactions between them; nodes and edges)? Please provide a description.

- The dataset consists of images depicting natural world phenomena (i.e., plant and animals species). In addition, it also contains natural language text queries representing scientific questions of interest. Each query is associated with a set of relevant images which came up after comprehensive labeling among the natural world image collection.

**How many instances are there in total (of each type, if appropriate)?**

- INQUIRE contains 200 text queries and a total of 24,336 relevant image matches.
- iNat24 contains 4,813,543 images from 9,959 species categories.

**Does the dataset contain all possible instances or is it a sample (not necessarily random) of instances from a larger set?**  If the dataset is a sample, then what is the larger set? Is the sample representative of the larger set (e.g., geographic coverage)? If so, please describe how this representativeness was validated/verified. If it is not representative of the larger set, please describe why not (e.g., to cover a more diverse range of instances, because instances were withheld or unavailable).

- The dataset contains approximately five million images sourced from iNaturalist. This is a subset of the total number of images present on iNaturalist. The selection and filtering process used to construct the dataset is described in Section H.

**What data does each instance consist of?**  "Raw" data (e.g., unprocessed text or images) or features? In either case, please provide a description.

- Each INQUIRE instance consists of a text query and a set of images representing all relevant matches for the query within iNat24.
- Each iNat24 instance is an image that is associated with a set of metadata, including the species label, location (latitude and longitude), observation time, license, image dimensions, and full taxonomic classification.

**Is there a label or target associated with each instance?**  If so, please provide a description.

- In INQUIRE each query is paired with a set of positive image matches from iNat24.
- iNat24 has species labels associated with each image. The species labels are obtained from 'research grade' labels that have been generated from the community consensus on iNaturalist.

**Is any information missing from individual instances?**  If so, please provide a description, explaining why this information is missing (e.g., because it was unavailable). This does not include intentionally removed information, but might include, e.g., redacted text.

- There is no information relevant to the task of the dataset omitted.

**Are relationships between individual instances made explicit (e.g., users' movie ratings, social network links)?**  If so, please describe how these relationships are made explicit.

- The image id, species taxonomy, locations, and time captured are provided with each image.

**Are there recommended data splits (e.g., training, development/validation, testing)?**  If so, please provide a description of these splits, explaining the rationale behind them.

- For INQUIRE, the queries and their relevant images are utilized solely for evaluation purposes within this paper and thus, there are no splits provided.
- The iNat24 dataset provides additional training data which can be used in conjunction with the validation and test splits from iNat21 [70]. More discussion of splits can be found in Section H.1.

**Are there any errors, sources of noise, or redundancies in the dataset?**  If so, please provide a description.

- While the species labels for each image in iNat24 are generated via consensus from multiple citizen scientists, there may still be errors in the labels which our evaluation will inherit. However, this error rate is estimated to be low [47].
- INQUIRE annotations may also contains noise in relevance scoring due to labeling error. However, we extensively labeled relevant queries to ensure this error rate is low.

**Is the dataset self-contained, or does it link to or otherwise rely on external resources (e.g., websites, tweets, other datasets)?**  If it links to or relies on external resources, a) are there guarantees that they will exist, and remain constant, over time; b) are there official archival versions of the complete dataset (i.e., including the external resources as they existed at the time the dataset was created); c) are there any restrictions (e.g., licenses, fees) associated with any of the external resources that might apply to a future user? Please provide descriptions of all external resources and any restrictions associated with them, as well as links or other access points, as appropriate.

- INQUIRE and iNat24 are self-contained datasets, as they include images and metadata that are directly available to download in their raw format without linking to any other external resources.

**Does the dataset contain data that might be considered confidential (e.g., data that is protected by legal privilege or by doctor-patient confidentiality, data that includes the content of individuals' non-public communications)?** If so, please provide a description.

- No, our dataset does not contain confidential data. The images that are part of it have been made publicly available by the users of iNaturalist.

**Does the dataset contain data that, if viewed directly, might be offensive, insulting, threatening, or might otherwise cause anxiety?** If so, please describe why.

- iNat24 contains pictures of the natural world (e.g., plant and animal species) captured by community volunteers. Some natural world images in this dataset could be disturbing to some viewers, e.g., there are a small number of images that contain dead animals. We include these images in the dataset as they are ecologically and scientifically useful, e.g., for studying the impact of roadkill on animal populations.

**Does the dataset relate to people?** If not, you may skip the remaining questions in this section.

- No, our dataset does not relate directly to people. Images of humans where their faces are visible have been filtered out using a combined manual and automated process. See Section H.1 for a discussion of data filtering.

**Does the dataset identify any subpopulations (e.g., by age, gender)?** If so, please describe how these subpopulations are identified and provide a description of their respective distributions within the dataset.

- No, our dataset does not identify any human subpopulations.

**Is it possible to identify individuals (i.e., one or more natural persons), either directly or indirectly (i.e., in combination with other data) from the dataset?** If so, please describe how.

- Some images publicly uploaded by users to the iNaturalist platform contain identifiable information, including pictures containing human faces, IDs, or license plates. To address this, we filter iNat24 to remove all such instances that we can identify, including by running detection algorithms to find all instances of human faces. More details are provided in Section H.1.

- All photos used to construct iNat24 come from observations captured by community volunteers who have given their images a suitable license for research use. We respect these licenses by providing the license information for each image as well as the rights holder in the metadata. The user-provided rights holder name can contain the user's iNaturalist user ID. This information is already available publicly from the iNaturalist platform.

**Does the dataset contain data that might be considered sensitive in any way (e.g., data that reveals racial or ethnic origins, sexual orientations, religious beliefs, political opinions or union memberships, or locations; financial or health data; biometric or genetic data; forms of government identification, such as social security numbers; criminal history)?** If so, please provide a description.

- No, our dataset does not aim to contain any data that can be considered sensitive in the ways discussed above. Details on how we filter iNat24 to remove all sensitive data are provided in Section H.1.

**Any other comments?**

- N/A

### K.3 Collection

**How was the data associated with each instance acquired?** Was the data directly observable (e.g., raw text, movie ratings), reported by subjects (e.g., survey responses), or indirectly inferred/derived from other data (e.g., part-of-speech tags, model-based guesses for age or language)? If data was reported by subjects or indirectly inferred/derived from other data, was the data validated/verified? If so, please describe how.

- The queries contained within INQUIRE come from discussions and interviews with a range of experts including ecologists, biologists, ornithologists, entomologists, oceanographers, and forestry experts. This resulted in 200 text queries. Annotators were instructed to label candidate images from iNat24 as either *relevant* (i.e., positive match) or *not relevant* (i.e., negative match) to a query, and to mark an image as not relevant if there was reasonable doubt. To allow for comprehensive labeling, where applicable, iNat24 species labels were used to narrow down the search to a sufficiently small size to label all relevant images for the query of interest. The annotation process is outlined in Section H.2.

**Over what timeframe was the data collected?** Does this timeframe match the creation timeframe of the data associated with the instances (e.g., recent crawl of old news articles)? If not, please describe the timeframe in which the data associated with the instances was created. Finally, list when the dataset was first published.

- The collection of iNat24 started with a iNaturalist observation database export generated on 2023-12-30. From this export, we filter observations to only include those added to iNaturalist in the years 2021, 2022, or 2023.
- The collection of INQUIRE queries and comprehensive labeling of their relevant images within iNat24 took place between January 2024 (following data export from iNaturalist) and end of May 2024.
- The dataset is not yet public, but will be made available prior to the NeurIPS 2024 conference conditioned on acceptance.

**What mechanisms or procedures were used to collect the data (e.g., hardware apparatus or sensor, manual human curation, software program, software API)?** How were these mechanisms or procedures validated?

- The iNat24 dataset was sourced from a GBIF export of the iNaturalist database.
- To comprehensively label the images that match each query in INQUIRE, we utilized a custom interface. For more information see Section H.2.

**What was the resource cost of collecting the data?**

- N/A

**If the dataset is a sample from a larger set, what was the sampling strategy (e.g., deterministic, probabilistic with specific sampling probabilities)?**

- iNat24 was sampled from an export of the iNaturalist platform and consists of image observations made in the years 2021, 2022, or 2023. Details about the sampling strategy can be found in Section H.1.

**Who was involved in the data collection process (e.g., students, crowdworkers, contractors) and how were they compensated (e.g., how much were crowdworkers paid)?**

- The queries contained within INQUIRE came from discussions and interviews with a range of experts including ecologists, biologists, ornithologists, entomologists, oceanographers, and forestry experts. Image annotation was performed by a carefully selected team of paid

MSc students or equivalent, many with expertise in ecology allowing for labeling of difficult queries. These annotators were paid at the equivalent of $15.50 per hour.

**Were any ethical review processes conducted (e.g., by an institutional review board)?** If so, please provide a description of these review processes, including the outcomes, as well as a link or other access point to any supporting documentation.

- We received internal ethical approval for our query collection and data labeling (Edinburgh Informatics Ethics Review Panel 951781 and MIT Committee on the Use of Humans as Experimental Subjects Protocol 2404001276).

**Does the dataset relate to people?** If not, you may skip the remainder of the questions in this section.

- N/A. The dataset contains images of different plant and animal species that have been made publicly available by users of the citizen science platform iNaturalist under a creative commons or similar license.

**Did you collect the data from the individuals in question directly, or obtain it via third parties or other sources (e.g., websites)?**

- N/A

**Were the individuals in question notified about the data collection?** If so, please describe (or show with screenshots or other information) how notice was provided, and provide a link or other access point to, or otherwise reproduce, the exact language of the notification itself.

- N/A

**Did the individuals in question consent to the collection and use of their data?** If so, please describe (or show with screenshots or other information) how consent was requested and provided, and provide a link or other access point to, or otherwise reproduce, the exact language to which the individuals consented.

- N/A

**If consent was obtained, were the consenting individuals provided with a mechanism to revoke their consent in the future or for certain uses?** If so, please provide a description, as well as a link or other access point to the mechanism (if appropriate)

- N/A

**Has an analysis of the potential impact of the dataset and its use on data subjects (e.g., a data protection impact analysis)been conducted?** If so, please provide a description of this analysis, including the outcomes, as well as a link or other access point to any supporting documentation.

- N/A

**Any other comments?**

- N/A

### K.4 PREPROCESSING / CLEANING / LABELING

**Was any preprocessing/cleaning/labeling of the data done(e.g.,discretization or bucketing, tokenization, part-of-speech tagging, SIFT feature extraction, removal of instances, processing of missing values)?** If so, please provide a description. If not, you may skip the remainder of the questions in this section.

- Besides resizing, we do not modify the images. Data cleaning is done to remove personally identifiable information or otherwise unsuitable images.

**Was the "raw" data saved in addition to the preprocessed/cleaned/labeled data (e.g., to support unanticipated future uses)?** If so, please provide a link or other access point to the "raw" data.

- N/A

**Is the software used to preprocess/clean/label the instances available?** If so, please provide a link or other access point.

- We use the following to aid in preprocessing:
  - img2dataset: https://github.com/rom1504/img2dataset
  - OpenCLIP: https://github.com/mlfoundations/open_clip
  - Face detector: https://github.com/biubug6/Pytorch_Retinaface

**Any other comments?**

- N/A

## K.5   USES

**Has the dataset been used for any tasks already?** If so, please provide a description.

- In our paper we use the INQUIRE dataset to benchmark several multimodal models on text-to-image retrieval. It has not been used for any tasks prior to this.

**Is there a repository that links to any or all papers or systems that use the dataset?** If so, please provide a link or other access point.

- Currently there is no such repository as the dataset is not public. We will collate one after the dataset has been released.

**What (other) tasks could the dataset be used for?**

- The iNat24 dataset could be used for training supervised fine-grained image classifiers. It could also be used for training self-supervised methods. The text pairs in INQUIRE could potentially be used to fine-tune fine-grained image generation models and vision language models.

**Is there anything about the composition of the dataset or the way it was collected and preprocessed/cleaned/labeled that might impact future uses?** For example, is there anything that a future user might need to know to avoid uses that could result in unfair treatment of individuals or groups (e.g., stereotyping, quality of service issues) or other undesirable harms (e.g., financial harms, legal risks) If so, please provide a description. Is there anything a future user could do to mitigate these undesirable harms?

- The images from the iNat24 dataset are not uniformly distributed across the globe (see Figure A3). Their spatial distribution reflects the spatial biases present in the iNaturalist platform. As a result, image classifiers trained on these models may preform worse on images from currently underrepresented regions.
- To decrease this bias we sample from spatio-temporal clusters of "observations groups". Observation groups are formed by grouping observations together if they are observed on the same day within 10km of each other, regardless of the observer. When sampling observations for a species, we cluster their associated observation groups using a spatio-temporal distance metric and then sample one observation per cluster in a round-robin fashion until we hit a desired sample size. When sampling within a cluster, we prioritize novel observation groups and novel users.

- In future, this issue could be further mitigated as more data from currently underrepresented regions becomes available.

**Are there tasks for which the dataset should not be used?** If so, please provide a description.

- There could be unintended negative consequences if conservation assessments were made based on the predictions from biased or inaccurate models developed to perform well on INQUIRE. Where relevant, we have attempted to flag these performance deficiencies in the main paper.

- While we have filtered out personally identifiable information from our images, the retrieval paradigm allows for free-form text search. In real-world text-to-image retrieval applications care should be taken to ensure that appropriate text filters are in-place to prevent inaccurate or hurtful associations being made between user queries and images of wildlife.

**Any other comments?**

- N/A

## K.6  DISTRIBUTION

**Will the dataset be distributed to third parties outside of the entity (e.g., company, institution, organization) on behalf of which the dataset was created?** If so, please provide a description.

- Yes, INQUIRE and iNat24 will be publicly available for download.

**How will the dataset will be distributed (e.g., tarball on website, API, GitHub)?** Does the dataset have a digital object identifier (DOI)?

- The dataset is distributed as a tarball. Links to the dataset download are available on our GitHub repository at https://github.com/inquire-benchmark/INQUIRE

**When will the dataset be distributed?**

- The dataset will be publicly released conditioned on acceptance.

**Will the dataset be distributed under a copyright or other intellectual property (IP) license, and/or under applicable terms of use (ToU)?** If so, please describe this license and/or ToU, and provide a link or other access point to, or otherwise reproduce, any relevant licensing terms or ToU, as well as any fees associated with these restrictions.

- The dataset will have the following ToU: By downloading this dataset you agree to the following terms
  - You will abide by the iNaturalist Terms of Service https://www.inaturalist.org/pages/terms.
  - You will use the data only for non-commercial research and educational purposes.
  - You will NOT distribute the dataset images.
  - The University of Massachusetts Amherst makes no representations or warranties regarding the data, including but not limited to warranties of non-infringement or fitness for a particular purpose.
  - You accept full responsibility for your use of the data and shall defend and indemnify the University of Massachusetts Amherst, including its employees, officers and agents, against any and all claims arising from your use of the data, including but not limited to your use of any copies of copyrighted images that you may create from the data.

**Have any third parties imposed IP-based or other restrictions on the data associated with the instances?** If so, please describe these restrictions, and provide a link or other access point

to, or otherwise reproduce, any relevant licensing terms, as well as any fees associated with these restrictions.

- Each image is accompanied with a specific license selected by the contributor. See the dataset for details.

**Do any export controls or other regulatory restrictions apply to the dataset or to individual instances?** If so, please describe these restrictions, and provide a link or other access point to, or otherwise reproduce, any supporting documentation.

- N/A

**Any other comments?**

- N/A

## K.7 MAINTENANCE

**Who is supporting/hosting/maintaining the dataset?**

- The dataset is hosted on AWS supported by the AWS Open Data Program.

**How can the owner/curator/manager of the dataset be contacted (e.g., email address)?**

- Questions, clarifications, and issues can be raised via the GitHub page: [https://github.com/inquire-benchmark/INQUIRE](https://github.com/inquire-benchmark/INQUIRE)

**Is there an erratum?** If so, please provide a link or other access point.

- Issues can be raised via the GitHub page: [https://github.com/inquire-benchmark/INQUIRE](https://github.com/inquire-benchmark/INQUIRE)

**Will the dataset be updated (e.g., to correct labeling errors, add new instances, delete instances)?** If so, please describe how often, by whom, and how updates will be communicated to users (e.g., mailing list, GitHub)?

- There may be a future version of the dataset, however we do not intend for the dataset to be frequently changing.

**If the dataset relates to people, are there applicable limits on the retention of the data associated with the instances (e.g., were individuals in question told that their data would be retained for a fixed period of time and then deleted)?** If so, please describe these limits and explain how they will be enforced.

- N/A

**Will older versions of the dataset continue to be supported/hosted/maintained?** If so, please describe how. If not, please describe how its obsolescence will be communicated to users.

- Previous versions of the iNaturalist image datasets can be found here [https://github.com/visipedia/inat_comp/tree/master](https://github.com/visipedia/inat_comp/tree/master)

**If others want to extend/augment/build on/contribute to the dataset, is there a mechanism for them to do so?** If so, please provide a description. Will these contributions be validated/verified? If so, please describe how. If not, why not? Is there a process for communicating/distributing these contributions to other users? If so, please provide a description.

- Contributors can join the iNaturalist platform: [https://www.inaturalist.org/](https://www.inaturalist.org/)

1129 **Any other comments?**

1130 • N/A