# OpenReview forum: "INQUIRE: A Natural World Text-to-Image Retrieval Benchmark"
_NeurIPS.cc/2024/Datasets_and_Benchmarks_Track — NeurIPS 2024 Track Datasets and Benchmarks Poster_

### Official Review · Reviewer_aBR1 · 2024-07-10

**Rating:** 7
**Confidence:** 3
**Clarity:** The paper is clear, and easy to under…

**Review:**

Overall, the paper is well motivated, original, and easy to understand. This paper introduces a benchmark which targets a key problem in VLLM retrieval: retrieving over large sets of images. The proposed benchmark is capable of evaluating the task (albeit in a limited set of domains), and is expected to be of high quality given the annotation process. Speaking of annotation process, the data collection, labeling and selection process is comprehensively explained in the appendix, and it's clear that the authors spent a lot of time working on the transparency of the data collection. The results demonstrated in the paper are not surprising (most models are quite poor at large-scale retrieval), but are helpful for understanding how existing models can perform on the task.

There are some key weaknesses to the proposed benchmark:
- **Size**: The benchmark is fairly small on the query dimension, consisting of only 200 queries to 5M base images. While small benchmarks is not a particularly large issue (and it's expectedly small, given the annotation effort), it does make me wonder what the expected variance of the estimator is when it comes to model performance. It would be good to see some analysis of the variance leveraging a tool such as bootstrap sampling to estimate the variance in the performance for the models.
- **Metrics**: The metrics chosen for the benchmark are not immediately intuitive, and require quite a verbose explanation, which is relegated to Appendix G. The justification for avoiding Prec@k for full-rank retrieval is somewhat weak. The claim that "Since calculating AP@k requires both the relevance and position of the top k items, AP@k may be prefered over Precision at k (P@k) which does not use position" is a good point, but given that it appears that relevance has binary labeling in the dataset, it's not clear that a higher rank for certain relevant images is more important. Similar for the nDCG metric. It would be nice to discuss this in detail in the paper as opposed to the appendix, as well as potentially include the more intuitive Prec@k metric (how many retrieved images out of K are relevant on average) in the tables.
- **Discussion**: The results and discussion in Section 5 are fairly surface level, and most do not show particularly deep insights into existing retrieval models that are not already well known/well accepted principles (small models are worse, high quality data is important, etc.). It might be more interesting to dive deeper into the results on L255 (Reranking offers a valuable opportunity for improving retrieval), which is something that the paper is well positioned to evaluate, and of more general interest to readers as a take-away point. Another thing that would be interesting is an expansion of the discussion on L264 - is there a quantitative way of ranking query complexity? Do these metrics correlate with model performance? Given that there are only 200 queries, such queries could even be ranked by hand or categorized by hand into query types, and performance evaluated there.

Minor issues:
- Table placement for Table 2 is a bit weird, as it appears far before the results section of the paper.

**Strengths:**

See Review (Copied from above)

This paper introduces a benchmark which targets a key problem in VLLM retrieval: retrieving over large sets of images. The proposed benchmark is capable of evaluating the task (albeit in a limited set of domains), and is expected to be of high quality given the annotation process. Speaking of annotation process, the data collection, labeling and selection process is comprehensively explained in the appendix, and it's clear that the authors spent a lot of time working on the transparency of the data collection. The results demonstrated in the paper are not surprising (most models are quite poor at large-scale retrieval), but are helpful for understanding how existing models can perform on the task.

**Additional Feedback:**

N/A

**Correctness:**

The claims appear to be correct, and sound. The paper does claim however that "Our evaluations use pre-trained models evaluated deterministically, thus we cannot include error bars", however such error bars could be computed using variance estimation techniques such as bootstrap sampling, and would be helpful to the understanding in the paper.

**Documentation:**

Dataset documentation is clear, and detailed.

**Ethics:**

No concerns.

**Limitations:**

While the limitations are discussed, many of the discussed limitations are in the Appendix, and could be gathered in Section 6.

**Opportunities For Improvement:**

See Review (Copied from above)

There are some key weaknesses to the proposed benchmark:
- **Size**: The benchmark is fairly small on the query dimension, consisting of only 200 queries to 5M base images. While small benchmarks is not a particularly large issue (and it's expectedly small, given the annotation effort), it does make me wonder what the expected variance of the estimator is when it comes to model performance. It would be good to see some analysis of the variance leveraging a tool such as bootstrap sampling to estimate the variance in the performance for the models.
- **Metrics**: The metrics chosen for the benchmark are not immediately intuitive, and require quite a verbose explanation, which is relegated to Appendix G. The justification for avoiding Prec@k for full-rank retrieval is somewhat weak. The claim that "Since calculating AP@k requires both the relevance and position of the top k items, AP@k may be prefered over Precision at k (P@k) which does not use position" is a good point, but given that it appears that relevance has binary labeling in the dataset, it's not clear that a higher rank for certain relevant images is more important. Similar for the nDCG metric. It would be nice to discuss this in detail in the paper as opposed to the appendix, as well as potentially include the more intuitive Prec@k metric (how many retrieved images out of K are relevant on average) in the tables.
- **Discussion**: The results and discussion in Section 5 are fairly surface level, and most do not show particularly deep insights into existing retrieval models that are not already well known/well accepted principles (small models are worse, high quality data is important, etc.). It might be more interesting to dive deeper into the results on L255 (Reranking offers a valuable opportunity for improving retrieval), which is something that the paper is well positioned to evaluate, and of more general interest to readers as a take-away point. Another thing that would be interesting is an expansion of the discussion on L264 - is there a quantitative way of ranking query complexity? Do these metrics correlate with model performance? Given that there are only 200 queries, such queries could even be ranked by hand or categorized by hand into query types, and performance evaluated there.

Minor issues:
- Table placement for Table 2 is a bit weird, as it appears far before the results section of the paper.

**Relation To Prior Work:**

While the set of related image retrieval datasets mentioned in the paper is fairly limited, and no models are compared across datasets, the contribution is sufficient.

**Summary And Contributions:**

This paper introduces the INQUIRE benchmark, a set of 200 queries over the iNat24 benchmark (5M images) for which relevance is labeled. The paper further provides evaluation of several models across two key challenges in image retrieval: (1) full-rank retrieval where models are expected to identify relevant images across all 5M samples, and (2) re-ranking, where models are expected to rank relevant images across 50/100 top retrieved samples.

---

> ### Author Rebuttal · Authors · 2024-08-17
>
> **[aBR1-1] Dataset size.**
> While INQUIRE contains 200 queries, each text query has *multiple* images associated with it. Specifically, there are 24,336 images labeled as relevant out of 149,022 total that were annotated. If we tabulated the size of our dataset in a similar fashion to other text-to-image retrieval datasets who report size in terms of the number of text-image pairs, this would correspond to 24,336 individual queries. Since the submission deadline, we have also added an additional validation set (see response [gN6o-2] to reviewer **gN6o**), which brings the total number of queries to 242, and the number of relevant images to 29,867.
>
> **[aBR1-2] Estimate of the variance of the performance of different models.**
> We have calculated the margin of error (95% CI) and report the results for several models below. We see that the margin of error is often smaller than the differences in performance between classes of models and do not impact our main findings. However, we think that this method for calculating variance is an overestimate of the true margin of error. Typically, variance estimation treats each sample as a scalar value outcome (e.g. in classification, is the model prediction correct or not correct?). However, our image-to-text retrieval task is complex, with each metric computed over 50 retrieved and ranked images. Thus, each individual task holds significantly more predictive power than is reflected by reducing its outcome to a single scalar value. We are not aware of existing techniques for accurately capturing the variability of models in such a setting, and think this is an interesting direction for further exploration.
>
> |Model|PR@50|mAP@50|nDCG@50|MRR|
> |-|-|-|-|-|
> |wildclip|12.1 ± 2.4|7.3 ± 2.1|16.1 ± 2.7|0.33 ± 0.05|
> |bioclip|6.2 ± 2.2|3.6 ± 1.8|7.1 ± 2.3|0.15 ± 0.04|
> |vit-b-16|15.8 ± 2.7|11.4 ± 2.5|22.7 ± 3.2|0.43 ± 0.06|
> |vit-l-14-dfn|28.1 ± 3.6|24.6 ± 3.7|39.3 ± 3.9|0.57 ± 0.06|
> |vit-h-14-378|36.6 ± 4.0|35.6 ± 4.0|51.8 ± 3.9|0.71 ± 0.05|
> |siglip-so400m-14-384|34.7 ± 3.9|34.8 ± 4.1|50.3 ± 4.1|0.70 ± 0.05|
>
> **[aBR1-3] Justification of evaluation metric used.**
> We agree that Precision@k provides an intuitive metric for evaluating full-rank retrieval, despite its weaknesses. We will make use of the extra page in the final camera ready text to move the discussion and justification of the metrics used from the appendix to the main paper. In addition, we will make the text more concise so that it is easier for the reader to understand. As suggested, we will also update the paper with a table that includes P@k for CLIP models (see table below).
>
> |Model|Precision@50|
> |-|-|
> |wildclip|12.1|
> |bioclip|6.2|
> |vit-b-16|15.8|
> |vit-l-14|20.4|
> |vit-l-14-dfn|28.1|
> |vit-h-14-378|36.6|
> |siglip-so400m-14-384|34.7|
>
> Regarding the justification for avoiding P@k, we would like to make the further point that since the denominator for precision is k, P@k can never reach a value of 1 when the total number of positive matches is fewer than k. This feature gives the undesirable property that even a perfectly performing model cannot achieve a perfect score.
>
> **[aBR1-4] Expanding the discussion of results.**
> We agree that the paper would benefit from additional discussion of the results. We plan to use the additional content page that is available for the camera ready version of the paper to expand on this discussion (e.g. further analysis of reranking performance). Understanding “ranking query complexity” is an interesting idea which we are investigating. We examined query complexity by studying queries that contain expert scientific terminology or lingo. Below, we have calculated the full-rank retrieval performance for queries that do and do not contain scientific lingo for a few CLIP models.
>
> ||Lingo|No Lingo|Lingo|No Lingo|
> |-|-|-|-|-|
> |Model|mAP@50|mAP@50|nDCG@50|nDCG@50|
> |wildclip-t1t7-lwf|5.8| 7.5|10.9|16.8|
> |bioclip|5.4|3.3|7.5|7|
> |vit-b-16|13.7|11|20|23.1|
> |vit-l-14-dfn|25|24.5|36.4|39.8|
> |vit-h-14-378|28.1|36.6|41.5|53.2|
> |siglip-so400m-14-384|28|35.8|39.5|51.8|
>
> We see that models generally struggle with queries that contain lingo. The top-performing models show a decrease of 8 points in mAP and 12 points in nDCG for queries with lingo. This result shows that difficult terminology is a weakness of current models and an opportunity for future methods to improve.
>
> As suggested, we will also move the placement of Table 2 so that it is closer to the results section.
>
> **[aBR1-5] Grouping the queries into different query types.**
> We have already manually grouped the queries into different “query types” based on their visual task category. Figure 2 illustrates our two level hierarchy, where we have the four coarse groups (Species, Context, Behavior, and Appearance) comprising 16 categories. The results by category are presented in Figure 5 and 6 with detailed results in Table A2 and A3. We will expand the discussion with this and further analysis, and we will update the final text to make it clearer how these types/groups were constructed and how the queries were assigned to them.
>
> **[aBR1-6] Moving some of the existing discussion of the limitations from the appendix to the main paper.**
> Given the extra space available for the camera ready version of the paper we will move some of the additional discussion of the limitations to section 6. This is an easy fix to make and we acknowledge that it is a good suggestion.
>
> **[aBR1-7] Evaluation on existing image retrieval datasets.**
> We only report results on our INQUIRE dataset over the two retrieval tasks. As noted on lines 92-23 performance on other retrieval datasets such as Flickr30k and COCO is already close to saturated. For those interested, existing work has already evaluated many of the same models we use on these existing retrieval datasets. A comprehensive benchmark is available on the OpenCLIP retrieval leaderboard: https://github.com/mlfoundations/open_clip/blob/main/docs/openclip_retrieval_results.csv.

---

> > ### Comment · Reviewer_aBR1 · 2024-09-01
> > **Response**
> >
> > Thanks for your response - I would love to see the included tables in the final version or appendix. I still believe this is a good paper - my score remains unchanged.

---

### Official Review · Reviewer_gN6o · 2024-07-24
**novel benchmark for nature oriented expert-level text to image retrieval**

**Rating:** 7
**Confidence:** 4
**Clarity:** yes

**Review:**

Quality: The paper provides a comprehensive dataset of images with a variety of species covered, a detailed set of annotations and labeling criteria to ensure high quality data in benchmarking, and two tasks for robust evaluation (full rank / rerank) of potential models.

Clarity: The paper is well structured / logically flows, with clear sections detailing the dataset, tasks, and evaluation methods. Multiple illustrations are provided that effectively illustrate aspects of the dataset and evaluation.

Originality: The paper appears to be a significant increase in difficulty in comparison to iNat21 in not just re framing problem from classification into retrieval/ranking, but also in the nature of their expert-level queries. This dataset should have significance to expert/scientific users.

Significance: Admittedly I do not work in the natural science so I can comment on the exact utility of these sorts of expert-level queries, but from my perspective this dataset does pose significant challenges to both advanced image understanding and has interesting challenges for both the initial and reranking phases.


Pros:

- Comprehensive high quality dataset with well defined evaluation tasks
- Benchmark illustrates significant gap in existing models
- Potential for significant scientific impact


Cons:

- Niche focus (natural world + expert oriented queries) may limit immediate cross-domain applicability
- Lack of explicit training/validation set may make measuring progress of new methods difficult to compare/analyze in future

**Strengths:**

Strengths

- A significant new dataset that in itself is likely a substantial resource for the scientific community
- A novel benchmark oriented around expert-level text to image retrieval and demonstrating difficulty by evaluating current SOTA models
- Nature of task can motivate advancement of multimodal models or more sophisticated retrieval systems to address these expert-level queries
- Detailed methodology and thorough analysis / commentary provided

**Additional Feedback:**

looks like a challenging dataset that should benefit practitioners in the natural science community, kudos! most feedback left in other prompts of this review.

**Correctness:**

Are the claims made in the submission correct? yes

If the submission is a dataset, it is constructed in a sound way? yes

If it is a benchmark, are the evaluation methods and experiment design appropriate and performed correctly? yes

**Documentation:**

For datasets, is there sufficient detail on data collection and organization, availability and maintenance, and ethical and responsible use? yes

For benchmarks, is there sufficient detail to support reproducibility? yes

**Ethics:**

Do you suspect there are any ethical concerns with the submission that warrant further discussion or review? no

**Limitations:**

Have the authors adequately addressed the limitations and potential negative societal impact of their work? yes

**Opportunities For Improvement:**

Opportunities / limitations

- scope of application is highly specialized on natural world images / that may limit its application or generalizability to other domains
- it would have been interesting to have a training sub set of these data or some noisy/less vigorously labeled but larger subset of data that researchers could train/fine tune against - for example it would have been interesting to see impact of fine tuning one of the CLIP variants on such a dataset to see what kind of gains one would get over the pre-trained setting as the provided comparisons over bioclip / wildclip were ultimately trained on different data (that is to say it is unclear to me if general purpose zero shot models would ever aim to cover these expert-level queries given its niche utility to a specific community and would thus more naturally fit under a fine tuned / specialized model setting)

**Relation To Prior Work:**

yes

**Summary And Contributions:**

This paper introduces INQUIRE, a new text to image retrieval benchmark and evaluates existing multimodal vision language models on text to image retrieval for full rank and reranking tasks over natural world images.

Main contributions:

- New iNat24 dataset of 5 million images, 200 expert-level retrieval queries, and 24,000 labeled true matches,
- Benchmarking tasks for core text to image retrieval, namely end to end / full ranking and reranking tasks
- an evaluation of existing multimodal models (both open weight and proprietary in form of GPT4V/4o) demonstrating gap of performance of SOTA models on this challenging dataset
- by focusing on expert-level retrieval that requires both advanced image understanding and domain expertise, the authors may help drive advancements in real scientific utility that go beyond the every day objects/captions seen in common t2i retrieval datasets

---

> ### Author Rebuttal · Authors · 2024-08-17
>
> Thank you for the helpful comments and suggestions!
>
> **[gN6o-1] The proposed dataset is focused only on natural world tasks which may limit cross domain applicability.**
> While the images in INQUIRE do indeed come from the natural world, our queries represent a wide breadth of scientific topics that were sourced from a large set of experts. Our diverse queries correspond to a variety of visual concepts that are present in other challenging domains.  For example, our queries require reasoning about fine-grained concepts, behavior, context, and interactions; which are challenges that appear in many other domains. For this reason, we believe that progress on INQUIRE will be indicative of multimodal model cross-domain performance. Moreover, current retrieval datasets like Flickr30k and COCO, that contain general topics, have shown to be too easy, with performance quickly saturating (e.g. BLIP-2 scores 98.9 on Flickr30K). Thus, there is a pressing need for a challenging large-scale retrieval benchmark.
>
> **[gN6o-2] Adding a validation set.**
> We agree that having explicit validation and test splits would be valuable for researchers in future. Between submitting the paper and this rebuttal phase we collected and annotated images for 42 additional new queries. These new queries span the space of different coarse groupings displayed in Fig 2 and they contain 5,532 additional positive images in total. From this we have constructed a dedicated validation set containing 42 queries. Note that this does not impact the results in the main paper as the methods evaluated do not require a validation set. We hope that this validation set will make it easier for future researchers that wish to fine-tune models to make progress on our dataset.
>
> Examples of of the text queries from our validation set are listed here:
> - *Caribou without palmate antlers*
> - *Red-capped Plover performing broken wing distraction*
> - *an oak gall produced by a spongy oak apple gall wasp*
> - *hognose snake playing dead*
> - *A case made by a bagworm larva*
>
> The ranking of methods on the new validation set broadly aligns with the test set.
>
> **[gN6o-3] Constructing a noisy training set.**
> This is a great suggestion! One of the benefits of our iNat24 dataset is that it contains a very large number of images (i.e. 5 million) that could be used for training/fine-tuning different potential retrieval models. While we believe this is out of scope for the current project, all the pieces are in place for future research to explore this idea. Weak annotations could be generated on this set using a combination of captioning models and the known species labels that we already provide. Our datasets will enable new research to address questions such as catastrophic forgetting when fine-tuning CLIP-style models, a challenging open problem.
>
> **[gN6o-4] Should we expect zero-shot models to perform well on these types of tasks?**
> Although many of our queries require domain knowledge, much of this information is widely available on the internet and thus reasonable for even zero-shot models to answer, given sufficient and high-quality training data. For example, the 10,000 species represented in our dataset are approximately some of the most common species (by number of observations on the iNaturalist platform). Many of our queries are readily solvable by non-expert humans with a brief explanation (see the examples in Appendix A, or the full list in Appendix J). These factors suggest that INQUIRE contains a combination of potentially solvable and challenging visual retrieval tasks that will test the current and next generation of vision-language models. However, we expect significant improvement can come from better methods of incorporating domain knowledge in model predictions.

---

### Official Review · Reviewer_efky · 2024-09-01
**Review of INQUIRE**

**Rating:** 4
**Confidence:** 4
**Clarity:** The paper is well-written and easy to…

**Review:**

Thanks to the authors for providing us with such a detailed introduction to the detailed dataset and benchmark. This paper focuses on high-quality natural data and natural image retrieval tasks, proposing a text-image retrieval benchmark and a massive natural dataset. The strengths of this paper are as follows:
1. The dataset introduced by this paper comprises a vast amount of high-quality data, with numerous human-annotated labels covering a wide range of species. This is invaluable for advancing research and identifying new research directions.
2. The data instances provided are all real photographs of various natural species, making the dataset both challenging and highly useful for building and evaluating expert models.
3. The paper offers detailed evaluation results across a diverse array of baseline methods, including open-source, closed, and newly proposed methods, facilitating clearer presentations and comparisons.
4. The benchmark is challenging enough and is very useful for text-image retrieval research.

However, I do have some questions:

1. The proposed iNat24 dataset contains 4,813,543 images (149,022 labeled) spanning 9,959 species, representing an impressive and substantial workload. However, the INQUIRE benchmark includes only 200 queries, and the number of species is also limited. While I understand that these queries are carefully designed and encompass a wide variety of natural animal behaviors, I question whether such a small query set is sufficiently challenging and whether it offers a comprehensive test for the models.
2. The performance of models on FULLRANK is relatively poor and has not yet reached a usable level. Could this be due to traditional VLMs like CLIP and LVLMs like GPT-4V being generative models trained on general data? Since the domain gap exists between the pre-train data of VLMs and iNat24. It might be better to test few-shot cases.
3. I failed to find the random choice performance of FULLRANK and human (expert) performance for both 2 tasks. Could you provide these results?

In summary, this paper introduces a large, detailed, and high-quality dataset that offers numerous research opportunities for computer vision and VLM design. However, the tasks and benchmarks presented are not particularly novel, and the task design is somewhat simplistic.

**Strengths:**

The strengths are basically as described in the "review" part.
1. The dataset introduced by this paper comprises a vast amount of high-quality data, with numerous human-annotated labels covering a wide range of species. This is invaluable for advancing research and identifying new research directions.
2. The data instances provided are all real photographs of various natural species, making the dataset both challenging and highly useful for building and evaluating expert models.
3. The paper offers detailed evaluation results across a diverse array of baseline methods, including open-source, closed, and newly proposed methods, facilitating clearer presentations and comparisons.
4. The benchmark is challenging enough and is very useful for text-image retrieval research.

**Additional Feedback:**

N/A

**Correctness:**

The claims are made correctly, and the construction procedure of the dataset, the evaluation methods of the benchmark, and the experiment methods of the benchmark have been correctly explained.

**Documentation:**

The detailed instructions for the proposed dataset have been clearly released, including the repository URL, dataset URL, datasheet details, detailed statistics, and annotation details.

**Limitations:**

1. The questioning method is relatively simplistic and does not adequately evaluate the model’s generalization capabilities.
2. While introducing a detailed and extensive dataset is a significant contribution, the benchmark and task design lack sufficient novelty.
3. Although the dataset is vast and comprehensive, the queries and species covered by the benchmark are limited.

Other limitations are essentially as described in the “review” section.

**Opportunities For Improvement:**

The opportunities for improvement are basically as described in the "review" part.
Above that, compared to the text-image retrieval task, maybe the captions and the high-quality image-text pairs can be used to construct expert models or can be used for some interesting research and exploration.

**Relation To Prior Work:**

The relation to prior work is discussed.

**Summary And Contributions:**

The authors introduce a text-to-image retrieval benchmark for evaluating multimodal vision-language models and a detailed iNat24 dataset containing massive natural images. The retrieval benchmark evaluates VLMs with the FULLRANK and RERANK tasks. The benchmark includes the iNat24 dataset, which comprises 24000 total matches and a total number of 200 queries. The contributions of INQUIRE can be summarized as follows:
1. This paper introduced a text-image retrieval benchmark
2. This paper introduced a detailed and massive natural dataset i.e. iNat24, which provides more research and improves chances by releasing the massive dataset.
3. This paper focuses on popular retrieval problems and proposes 2 challenging tasks.

---

> ### Author Response · Authors · 2024-09-01
>
> Greetings **efky**,
>
>
> It looks like this is the same text as was shared yesterday. You can find our response appended as a rebuttal to your previous comment titled "Review of INQUIRE".

---

### Author Rebuttal · Authors · 2024-08-17

We thank the reviewers for their constructive comments. We have addressed individual responses below. We note that we currently only have two reviewers. We are unsure if we will receive a third review, so we will wait to upload a full general response here until then.

---

### Comment · Reviewer_efky · 2024-08-31
**Review of INQUIRE**

**Summary And Contributions:**

The authors introduce a text-to-image retrieval benchmark for evaluating multimodal vision-language models and a detailed iNat24 dataset containing massive natural images. The retrieval benchmark evaluates VLMs with the FULLRANK and RERANK tasks. The benchmark includes the iNat24 dataset, which comprises 24000 total matches and a total number of 200 queries. The contributions of INQUIRE can be summarized as follows:
1. This paper introduced a text-image retrieval benchmark
2. This paper introduced a detailed and massive natural dataset i.e. iNat24, which provides more research and improves chances by releasing the massive dataset.
3. This paper focuses on popular retrieval problems and proposes 2 challenging tasks.

**Review:**

Thanks to the authors for providing us with such a detailed introduction to the detailed dataset and benchmark. This paper focuses on high-quality natural data and natural image retrieval tasks, proposing a text-image retrieval benchmark and a massive natural dataset. The strengths of this paper are as follows:
1. The dataset introduced by this paper comprises a vast amount of high-quality data, with numerous human-annotated labels covering a wide range of species. This is invaluable for advancing research and identifying new research directions.
2. The data instances provided are all real photographs of various natural species, making the dataset both challenging and highly useful for building and evaluating expert models.
3. The paper offers detailed evaluation results across a diverse array of baseline methods, including open-source, closed, and newly proposed methods, facilitating clearer presentations and comparisons.
4. The benchmark is challenging enough and is very useful for text-image retrieval research.

However, I do have some questions:

1. The proposed iNat24 dataset contains 4,813,543 images (149,022 labeled) spanning 9,959 species, representing an impressive and substantial workload. However, the INQUIRE benchmark includes only 200 queries, and the number of species is also limited. While I understand that these queries are carefully designed and encompass a wide variety of natural animal behaviors, I question whether such a small query set is sufficiently challenging and whether it offers a comprehensive test for the models.
2. The RERANK performance of models shown in Table 4 is significantly higher than the two-stage FULLRANK performance presented in Table 3. Why is there such a large gap between these two tasks? Could generative models like GPT-4, VILA, or LLaVA 1.6 be directly applied to FULLRANK tasks?
3. The performance of models on FULLRANK is relatively poor and has not yet reached a usable level. Could this be due to traditional VLMs like CLIP and LVLMs like GPT-4V being generative models trained on general data? I wonder if few-shot instruction tuning or adjustments could improve their performance.
4. I failed to find the random choice performance and human (expert) performance for these benchmark tasks. Could you provide these results?

In summary, this paper introduces a large, detailed, and high-quality dataset that offers numerous research opportunities for computer vision and VLM design. However, the tasks and benchmarks presented are not particularly novel, and the task design is somewhat simplistic.

**Strengths:**

The strengths are basically as described in the "review" part.
1. The dataset introduced by this paper comprises a vast amount of high-quality data, with numerous human-annotated labels covering a wide range of species. This is invaluable for advancing research and identifying new research directions.
2. The data instances provided are all real photographs of various natural species, making the dataset both challenging and highly useful for building and evaluating expert models.
3. The paper offers detailed evaluation results across a diverse array of baseline methods, including open-source, closed, and newly proposed methods, facilitating clearer presentations and comparisons.
4. The benchmark is challenging enough and is very useful for text-image retrieval research.

**Rating:** 4 OK but Not good enough, reject

**Opportunities For Improvement:**

The opportunities for improvement are basically as described in the "review" part.
Above that, compared to the text-image retrieval task, maybe the captions and the high-quality image-text pairs can be used to construct expert models or can be used for some interesting research and exploration.

**Confidence**:  4: The reviewer is confident but not absolutely certain that the evaluation is correct

---

> ### Author Rebuttal · Authors · 2024-08-31
>
> Thanks for the helpful comments and suggestions. We are happy to see that you believe INQUIRE is “both challenging and highly useful for building and evaluating expert models”. We have responded to  your comments below. However, please do not hesitate to provide any follow up comments if there is anything that remains unclear as we will aim to reply before the discussion period ends tomorrow.
>
> **[efky-1] “While introducing a detailed and extensive dataset is a significant contribution, the benchmark lacks sufficient novelty.”**
> Respectfully, we disagree. We believe that INQUIRE is novel in several unique and important ways: (i) INQUIRE is the first multi-image text-to-image retrieval benchmark and (ii) INQUIRE is the first expert-level retrieval benchmark. INQUIRE is uniquely focused on the broad topic of assisting scientific discovery where progress can directly lead to better tools for scientists. Importantly, the images and task annotations that comprise INQUIRE are directly available to researchers via a convenient download link. We do not believe that there is another dataset that is similar in terms of challenge, size, accessibility, and annotation detail.
>
>
> **[efky-2] “Task design is somewhat simplistic”.**
> We confess that we are slightly confused by this comment as no justification is provided for why the reviewer believes this. If possible, it would be great if the reviewer could provide  further clarification so that we can address this question properly. We posit that our query design and selection is unique and contains real world challenging queries that have been sourced via extensive interactions with domain expert scientists and via literature reviews (see Section 3.2).  Additionally, INQUIRE is the first ever image retrieval benchmark to use information retrieval metrics such as nDCG, which is only possible by our multi-image benchmark design. We also introduce both Full-rank and Re-rank tasks which are designed for different stages of retrieval.
>
>
> **[efky-3] Is the dataset sufficiently challenging?**
> Yes, INQUIRE is a difficult expert-level benchmark that challenges state-of-the-art models. On the full-rank task (see Table 2), even the top models fail to score over 50 points in mAP, showing the difficulty of the expert-level queries in our dataset. Unlike previous text-to-image retrieval benchmarks where performance has saturated (see Lines 92-93), INQUIRE indicates that there is much-needed room for improvement.
>
>
> **[efky-4] Is INQUIRE a comprehensive test?**
> This is a good point. While INQUIRE may not comprehensively test every capability of a model, our queries cover a wide variety of topics (see Figure 2) requiring nuanced understanding of images at many levels. Thus, we believe that models that do well on INQUIRE necessarily have good general capabilities. Indeed, our results show that the best performing models are those with the best generalist capabilities (L249-258).
>
>
> **[efky-5] “Number of queries or species is limited.”**
> While INQUIRE contains 200 queries, each query has *many* images associated with it. Specifically, there are 24,336 images labeled as relevant. If we tabulated dataset size similarly to other text-to-image retrieval datasets who report size in terms of the number of text-image pairs, we would have 24,336 individual labels. Furthermore, as discussed in our response to **gN6o** since the initial paper deadline we have increased the number of queries to 242.
>
> Our number of species is quite large, as 200 queries cover many more than 200 species. Many queries cover multiple species, and Figure 3 shows that the species we cover provide broad taxonomic coverage. Please see the Appendix for a full list of our queries.
>
>
> **[efky-6] Performance difference between the full-rank and rerank tasks.**
> As we mention in the description of the rerank task (Lines 191-200 and the description of Table 4), the rerank task metrics are calculated based solely on the fixed set of 100 images per task, disregarding any potential positives outside this set. Full-rank performance is calculated on the full 5 million image dataset, so any missed positives outside the top k are penalized. Apologies if this was not clear in the text, it will be an easy fix to update the text to clarify this.
>
>
> **[efky-7] Could models like GPT-4 be directly applied to the full-rank task?**
> As there are 5 million images, it would be computationally intractable (~2 weeks per query for LLaVA-34B on an A100) to apply these multi-billion parameters models to each of the images. This is why in practice, two-stage systems are used, and is why we introduce the reranking task.
>
>
> **[efky-8] Has full-rank performance yet reached a usable level?**
> We respectfully disagree that full-rank performance has not reached a usable level. In fact, our top model is able to surface at least relevant images in the top 50 for most queries, which is already immediately useful for scientists. In addition, future progress on INQUIRE by the machine learning community will directly benefit scientists.
>
>
> **[efky-9] Could few shot instruction tuning be used?**
> We agree that few-shot instruction tuning and other improvements could increase performance. It is out of scope for this datasets and benchmarks paper, but we hope that INQUIRE will enable researchers to develop such methods in the future.
>
>
> **[efky-10] What is random choice or human performance?**
> Actually, random choice performance is already included in Table 4 (top left row) and discussed in Section 5.2. We do not include random choice performance for the full-rank task since the full dataset is sufficiently large that the random performance would be 0. While we do not have human performance metrics, we find for our tasks that humans are very consistent in visual labeling, with error significantly lower than any current models.

---

### Comment · Reviewer_efky · 2024-08-31
**Review of INQUIRE(Cont.)**

**Limitations:**

1. The questioning method is relatively simplistic and does not adequately evaluate the model’s generalization capabilities.
2. While introducing a detailed and extensive dataset is a significant contribution, the benchmark and task design lack sufficient novelty.
3. Although the dataset is vast and comprehensive, the queries and species covered by the benchmark are limited.

Other limitations are essentially as described in the “review” section.

**Correctness:**

The claims are made correctly, and the construction procedure of the dataset, the evaluation methods of the benchmark, and the experiment methods of the benchmark have been correctly explained.

**Clarity:**

The paper is well-written and easy to read.

**Relation To Prior Work:**

The relation to prior work is discussed.

**Documentation:**

The detailed instructions for the proposed dataset have been clearly released, including the repository URL, dataset URL, datasheet details, detailed statistics, and annotation details.

**Ethics:**

N/A

**Flag For Ethics Review:** 2: No, there are no or only very minor ethics concerns

**Additional Feedback:**

N/A

---

> ### Comment · Area_Chair_xoRQ · 2024-08-31
>
> Dear authors,
>
>      Could you provide your rebuttal to the review above? Thanks

---

> > ### Author Response · Authors · 2024-08-31
> > **Completed rebuttal to **efky**’.**
> >
> > We have completed the rebuttal to the review above, and we hope to discuss their feedback with the remaining time in the discussion period.

---

### Decision · Program_Chairs · 2024-09-26

**Decision:**

Accept (Poster)

**Comment:**

This is a hard paper to decide since there are similar papers which received similar ratings for text-to-image retrieval benchmarks. While the paper received a negative rating, the rating has not been backed up by the reviewer after rebuttal while the positive reviews contain strong points backing up the review and responded to the rebuttal. After having read the paper, I agree with the positive reviews. The dataset is important for domain expertise testing and I recommend an accept.